# MOBILEKGQA: ON-DEVICE KGQA SYSTEM ON DYNAMIC MOBILE ENVIRONMENTS

**Junyong Ahn**[1], **Hyeongrok Han**[2], **Bong Gyun Kang**[1], **Jisoo Mok**[3]
**Byunghan Lee**[4,†], **Sungroh Yoon**[1,2,†]
[1]Interdisciplinary Program in Artificial Intelligence, Seoul National University
[2]Department of ECE, Seoul National University
[3]Department of EE&CS, Daegu Gyeongbuk Institute of Science and Technology
[4]Department of EE, Seoul National University of Science and Technology
{tld215, gksgudpp, luckypanda, sryoon}@snu.ac.kr
jmok@dgist.ac.kr   bhlee@seoultech.ac.kr
[†]Corresponding author

## ABSTRACT

Developing a mobile system capable of generating responses based on stored user data is a crucial challenge. Since user data is stored in the form of Knowledge Graphs, the field of knowledge graph question answering (KGQA) presents a promising avenue towards addressing this problem. However, existing KGQA systems face two critical limitations that preclude their on-device deployment: resource constraints and the inability to handle data accumulation. Therefore, we propose MobileKGQA, the first on-device KGQA system capable of adapting to evolving databases with minimal resource demands. MobileKGQA significantly reduces computational overhead through embedding hashing. Moreover, it successfully adapts to evolving databases under resource constraints through a novel annotation generation method. Its mobile applicability is validated on the **NVIDIA Jetson Orin Nano edge-device platform**, achieving **20.3% higher performance while using only 30.4% of the energy** consumed by the SOTA (state-of-the-art). On standard KGQA benchmarks, using just **7.2% of the computation and 9% of the parameters**, MobileKGQA demonstrates performance that is empirically indistinguishable from the SOTA and outperforms baselines under distribution shift scenarios.

## 1 INTRODUCTION

In mobile devices, a vast amount of user information, such as their social interactions and activity logs, is stored in knowledge graph (KG) databases (Novović et al., 2017). These databases continuously accumulate new user information. By leveraging this user information in KG databases, on-device Large Language Models (LLMs) can deliver more customized responses to the user.

Knowledge Graph Question Answering (KGQA) (Sun et al., 2018; 2019) thus naturally rises as a promising approach to improve user experience of on-device LLMs through the integration of KG databases. KGQA aims to provide answers to queries by retrieving the information from KGs; recently, it has been shown to be effective at addressing limitations of LLMs, such as the difficulty in handling new information (Pan et al., 2024), unstable reasoning (Luo et al., 2024b), the lack of explainability (Luo et al., 2024c), and hallucinations (Wang et al., 2023a; Martino et al., 2023). Extending these advantages of KGQA systems to on-device LLMs enables the integration of the information stored in KG databases, such that on-device LLMs better align with the user's demands.

Although KGQA systems hold immense potential to enhance the quality of on-device LLMs, they face a critical limitation that precludes their deployment in mobile environments: poor adaptability to evolving KG databases. As users interact with their mobile devices, new information is continuously added to the KG databases (e.g., when emerging trends introduce new words or when a change of location updates location-based preferences). This continuous modification can induce substantial distribution shifts, and some studies have reported sharp shifts in real user data (Desiderio et al.,

Table 1: Limitations of various KGQA models with respect to mobile deployment. Favorable characteristics are highlighted in **red** (adaptability: adaptability to the distribution shift).

| category | model | computation | | adaptability |
|---|---|---|---|---|
| | | latency | tuning parameters | |
| w/o LLM | RnG-KBQA (Ye et al., 2022) | **low** | 443M | × |
| | DecAF (Yu et al., 2023) | **low** | 848M | × |
| w/ LLM (finetuned) | RoG (Luo et al., 2024c) | **low** | 7B | × |
| | ChatKBQA (Luo et al., 2024a) | **low** | 7B | × |
| w/ LLM (frozen) | KB-BINDER (Li et al., 2023a) | high | **-** | ○ |
| | ToG (Sun et al., 2024) | high | **-** | ○ |
| | **MobileKGQA** | **low** | **0.136M** | ○ |

2025; Miller et al., 2015). Moreover, these shifts have also been shown to cause performance degradation of KGQA systems over time (Dai et al., 2025) (also see Table 4). Therefore, current systems require frequent model retraining, which is typically performed on centralized servers, raising serious concerns for data privacy.

Given the challenges of adaptability and data privacy, developing a KGQA system that can be trained directly on mobile devices is a key research topic. However, no such system has been proposed (Table 1), and current mainstream approaches still rely on tuning billions of parameters. While some in-context learning (ICL)-based methods do not require training, they involve repeatedly invoking LLMs, resulting in high latency—making them unsuitable as well. To address this challenge, we propose MobileKGQA, the first KGQA system that can be deployed and trained directly on mobile devices. This work is driven by two key research questions: (1) How can we reduce the training resources required by current KGQA systems to a level feasible on edge devices? and (2) How can we generate annotations for supervised training without data leakage?

In response to the first question, MobileKGQA introduces a retrieval mechanism based on hash codes. This mechanism consists of two main components: the hashing module and the reasoning module. The hashing module significantly reduces computational cost by compressing high-dimensional, floating-point LLM embeddings into binary hash codes $\mathbf{h} \in \{-1, 1\}$. It achieves this by maximizing the mutual information between the original embedding and the hash code to preserve semantic content. As a result, unlike conventional approaches, MobileKGQA avoids the need to store gigabyte-scale embeddings for a few megabytes of text (Li et al., 2025) or to regenerate embeddings during training (Mavromatis & Karypis, 2025). The resulting hash codes are then passed to the reasoning module, which infers answer candidates with minimal computational and memory overhead. These candidates are used to retrieve the most plausible reasoning paths corresponding to the input question, allowing on-device LLMs to generate responses grounded in the structured knowledge stored in the KG database. By leveraging this hashing-based architecture, MobileKGQA dramatically reduces the computation and parameter count, paving the way for on-device adaptation.

To address the second research question—the absence of ground truth annotations (i.e., question-answer pairs) for newly accumulated knowledge—the system must generate annotations through complex reasoning over recently integrated information. Previous approaches have attempted to solve this problem by employing multiple LLM invocations (Jiang et al., 2023b) or relying on cloud-based LLMs (Li et al., 2024). However, such strategies are unsuitable for mobile environments due to constraints in computational resources and concerns over data privacy. MobileKGQA overcomes these limitations through a novel annotation generation process based on sequential reasoning. This method incrementally integrates structured knowledge to construct logically coherent and semantically consistent questions. By decomposing complex reasoning into simpler, interpretable steps, it not only enhances the quality of generated annotations but also significantly reduces the number of output tokens required, making it well-suited for resource-constrained mobile settings.

To thoroughly evaluate our proposed MobileKGQA model, we conducted experiments on the NVIDIA Jetson Orin Nano platform, which closely approximates a real mobile environment. The results show that MobileKGQA is the only model whose optimal configuration remains feasible on mobile devices, delivering 20.3% higher performance while consuming only 30.4% of the energy. In addition, evaluations on two standard KGQA benchmarks demonstrate that MobileKGQA achieves performance comparable to the latest state-of-the-art lightweight models while requiring just 7.2% of the computation and 9% of the parameters. Moreover, it exhibits superior adaptability to evolving data, consistently outperforming all baselines.

## 2 RELATED WORK

### 2.1 KNOWLEDGE GRAPH QUESTION ANSWERING

Conventional KGQA research mainly falls into two categories (Luo et al. (2022)): Information Retrieval (IR)-based and Semantic Parsing (SP)-based methods. More recently, LLM-based approaches have emerged. Additional details not covered in this section are presented in Appendix B.

**IR-based methods** construct subgraphs from knowledge graphs to extract information relevant to a given question. Then, the model learns to reason about the answer using the retrieved graph and the question (Miller et al., 2016; Sun et al., 2019). Research has focused on improving the reasoning process. For example, NSM (He et al., 2021) applied a knowledge distillation method to provide the supervision signals in intermediate steps, while EmbedKGQA (Saxena et al., 2020) used graph embeddings to improve reasoning performance on sparse graphs with many missing links.

**SP-based methods** infer answers by converting given questions to query languages (*e.g.,* SPARQL) that are compatible with a knowledge graph. The query language is obtained either through query graph generation (Yih et al., 2016; Lan & Jiang, 2020; Jiang et al., 2023c) or by a seq2seq model (Shu et al., 2022; Yu et al., 2023). For instance, QGG (Lan & Jiang, 2020) generates a query subgraph by taking extend, connect, and aggregate actions at each step. On the other hand, DecAF (Yu et al., 2023) leverages a seq2seq model-based joint decoding to generate both query language and natural language answers.

**LLM-based methods** can be categorized into two groups. The first category employs In-context Learning (ICL), aiming to retrieve relevant data. It utilizes prompt engineering and CoT reasoning (Li et al., 2023a; Sun et al., 2024; Gu et al., 2023). The second category fine-tunes LLMs to generate query languages or reasoning paths from given queries. (Luo et al., 2024a;c).

### 2.2 KGQA FOR MOBILE ENVIRONMENT

Despite advances in KGQA systems, existing models face considerable challenges when deployed in mobile environments. IR-based models, for instance, require high computation as they need to store high-dimensional embeddings (Saxena et al., 2020) and require repetitive computations for subgraph construction (Zhang et al., 2022). On the other hand, SP-based models struggle with query language generation, often producing non-executable logical forms when new relations are introduced to the database (Xie et al., 2022; Zhang et al., 2023). Lastly, LLM-based models utilizing ICL exhibit high latency, driven by processes such as multiple CoT reasoning (Sun et al., 2024) or iterative logical form generation (Li et al., 2023a). Additionally, models requiring LLM fine-tuning encounter significant computational burdens, especially under distribution shifts (Luo et al., 2024a;c).

## 3 PRELIMINARY

**Knowledge Graphs.** A knowledge graph is defined as a set of triples $\mathcal{G} = \{(h, r, t) \mid h, t \in \mathcal{V}, r \in \mathcal{R}\}$, where $\mathcal{V}$ represents entity set and $\mathcal{R}$ represents the relation set. A **triple** $(h, r, t)$ represents a knowledge where relation $r$ exists between the head entity $h$ and the tail entity $t$.

**KGQA and Distribution Shift.** Knowledge Graph Question Answering (KGQA) is a task that aims to answer natural language queries using information stored in knowledge graphs. Formally, given a query $q$ and a knowledge graph $\mathcal{G}$, KGQA seeks to learn a function $f$ that generates the correct answer $a$:

$$a = f(q, \mathcal{G})$$

Following a scenario commonly occurring in mobile devices, where new data is gradually added over time, we define the distribution shift in knowledge graphs as the accumulation of new data from a different distribution:

$$\mathcal{G}_{\text{shifted}} = \mathcal{G}_{\text{ori}} \cup \mathcal{G}_{\text{new}} \quad \text{s.t.} \quad \mathcal{G}_{\text{ori}} \sim P_{\text{ori}} \ \& \ \mathcal{G}_{\text{new}} \sim P_{\text{new}}$$

where $\mathcal{G}_{\text{ori}}$ represents the original knowledge graph following original distribution $P_{\text{ori}}$, and $\mathcal{G}_{\text{new}}$ represents the newly introduced knowledge graph following different distribution $P_{\text{new}}$. The function

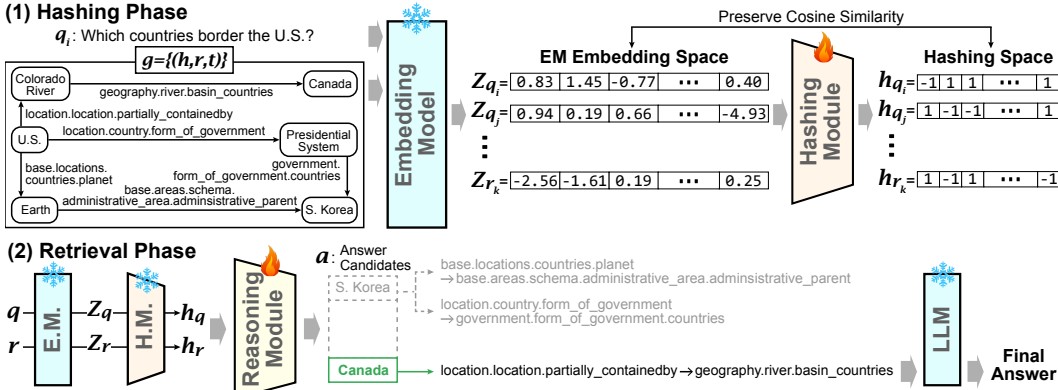

Figure 1: Pipeline of the hashing and retrieval phase: 1) The hashing module is trained to transform embeddings of questions and relations into binary hash codes while preserving their semantic information. 2) The reasoning module selects answer candidates and specifies query-relevant paths. 3) The LLM selects the optimal reasoning path from the retrieved paths and generates answers. E.M. and H.M. indicate embedding model and hashing module.

$f$ is learned using a dataset $\mathcal{D}_{\text{train}} = \{\{(q_i, a_i)_{\text{ori}}\}_{i=1}^M, \mathcal{G}_{\text{shifted}}\}$ and tested using a dataset $\mathcal{D}_{\text{test}} = \{\{(q_j, a_j)_{\text{shifted}}\}_{j=1}^N, \mathcal{G}_{\text{shifted}}\}$.

Following prior research (Wang et al., 2023b; Luo et al., 2024c), we refer **question entities** $e_q$ as words that are mentioned in the question, and assume that these entities are connected to answers through relations in the graph. Among the paths between question entities and answers, we refer **reasoning paths** as the paths that exhibit logical coherence with the given question.

## 4 METHODS

MobileKGQA consists of three phases: (1) a hashing phase that minimizes the computational cost of the reasoning module, (2) a retrieval phase that enables efficient data retrieval, and (3) an adaptation phase that addresses distribution shift, which operates when new data is introduced and the database is updated. The complete framework is illustrated in Figure 1 and 2. Our code is available at `https://github.com/jyahn215/mobileKGQA`.

### 4.1 HASHING PHASE

In the hashing phase, we project embeddings obtained from the embedding model (EM) into low-dimensional hash codes while retaining semantic information. To achieve this, we construct the embedding set $Z$ for the questions $\mathcal{Q}$ and the relations $\mathcal{R}$ of the knowledge graph.

$$Z = \{z \mid z = \text{EM}(x), x \in \mathcal{Q} \cup \mathcal{R}\} \tag{1}$$

Since hashing involves discrete optimization over binary codes, we approximate this transformation as a two-stage mapping for tractable optimization: a low-dimensional mapping $\phi$ followed by a binarization mapping $\psi$.

$$\text{Maximize}; I(z, h); \text{subject to}; h = \psi(\phi(z))$$

To implement this transformation, we apply a low-dimensional mapping $\phi$ using a multi-layer perceptron (MLP). Then, we use batch normalization (BN) to enforce zero-centered distributions, which promotes a more uniform spread in the hashing space and decreases the chances that different representations share identical codes.

$$d = \tanh(\text{BN}(\text{MLP}_\phi(z))), \quad h = \text{sgn}(d) \tag{2}$$

Based on the above implementation, we propose the following notation and assumption.

**Notation.** Let $\mathbf{z} \in \mathbb{R}^D$ be an embedding and $\mathbf{h} \in \{-1, 1\}^d$ a hash code. We express them in polar coordinates as $\mathbf{z} = (r_z, \boldsymbol{\theta_z})$ and $\mathbf{h} = (r_h, \boldsymbol{\theta_h})$, where $r$ denotes the norm and $\boldsymbol{\theta}$ the orientation.

**Assumption 1** (Independence Assumption). *The norm of embeddings $r_z$ is independent of both the orientation of embeddings $\boldsymbol{\theta_z}$ and the orientation of hash codes $\boldsymbol{\theta_h}$, i.e.,*

$$P(r_z, \boldsymbol{\theta_z}) = P(r_z)P(\boldsymbol{\theta_z}) \quad and \quad P(r_z, \boldsymbol{\theta_h}) = P(r_z)P(\boldsymbol{\theta_h}).$$

Given empirical studies of LLM representation (Pennington et al., 2014) and the feature-wise normalization of batch normalization, these assumptions provide a reasonable approximation. Building on this, we formally present the conditions for maximizing mutual information.

**Theorem 1** (Conditions for Mutual Information Maximization). *Suppose Assumption 1 holds, and both $\phi$ and $\psi$ preserve the cosine similarity between any pair of embeddings in Z. Then, the mutual information $I(z, h)$ is maximized.*

The proof of Theorem 1 is provided in Appendix C. To optimize our hashing module, we employ a log-ratio loss (Kim et al., 2019), which was originally proposed for regression problems in the computer vision domain. In our implementation, we establish the cosine similarities between embeddings before each mapping as regression targets, thereby training the mapping to preserve cosine similarity. Specifically, the loss function is formulated as follows, where $\alpha$ represents a hyperparameter to balance the two mappings, sim serves as a cosine similarity function, and $\ell$ denotes the log-ratio loss:

$$L_{\text{hash}} = \ell_\phi(Z_a, Z_i, Z_j) + \alpha\ell_\psi(d_a, d_i, d_j) \tag{3}$$

$$\ell_{\text{f}}(a, i, j) = \left\{ \log \frac{\text{sim}(f(a), f(i))}{\text{sim}(f(a), f(j))} - \log \frac{\text{sim}(a,i)}{\text{sim}(a,j)} \right\}^2 \tag{4}$$

## 4.2 RETRIEVAL PHASE

During the retrieval phase, the reasoning module predicts answer candidates from the knowledge graph. It then extracts the paths linking the query entities to these candidates, effectively restricting the range of plausible reasoning. The retrieved paths are subsequently provided to the LLM to produce the final answer.

**Reasoning Module** To implement an answer node selection process effectively, we implement a reasoning module based on a GNN, leveraging the ReaRev architecture (Mavromatis & Karypis, 2022) which effectively utilizes graph structure as an inductive bias. This reasoning module $RM$ takes three inputs – the query hash code $h_q$, the relation hash code $h_r$, and the graph $\mathcal{G}$ – and generates node representations $H$ of the graph.

$$H = \{h_i\}_{i=1}^N = RM(h_q, h_r, \mathcal{G}) \tag{5}$$

The reasoning module is trained through KL-divergence loss, with answer nodes as labels $Q$.

$$L_{\text{reason}} = D_{\text{KL}}(Q \,\|\, \text{softmax}(WH)) \tag{6}$$

Given query entities $e_q$ and answer candidates $\{\hat{a}_j\}_{j=1}^M$, we retrieve the shortest paths set $\mathcal{P}$ that links the $e_q$ to $\hat{a}_j$, and LLM generates the final answer based on these paths.

$$\mathcal{P} = \left\{ p^k := e_q^k \to r_0^k \to \cdots \to r_{l-1}^k \to \hat{a}_j^k \right\}_{k=1}^K$$

While each path $p^k$ captures meaningful relationships between the $e_q^k$ and $\hat{a}_j^k$, it may not necessarily align with the ground-truth reasoning path. However, prior studies (Li et al., 2025; Mavromatis & Karypis, 2025; Luo et al., 2024c) demonstrated that the shortest paths can serve as an effective heuristic. Moreover, given the exponential growth of the search space with path length, focusing on the shortest paths can be a practical way to optimize the retrieval budget.

## 4.3 ADAPTING TO DISTRIBUTION SHIFTS

In the adaptation phase, our annotation generation method enables efficient multi-hop reasoning by breaking complex reasoning into smaller steps. These substeps provide a clear structure for effective reasoning, reducing unnecessary token generation. The entire process is illustrated in Figure 2.

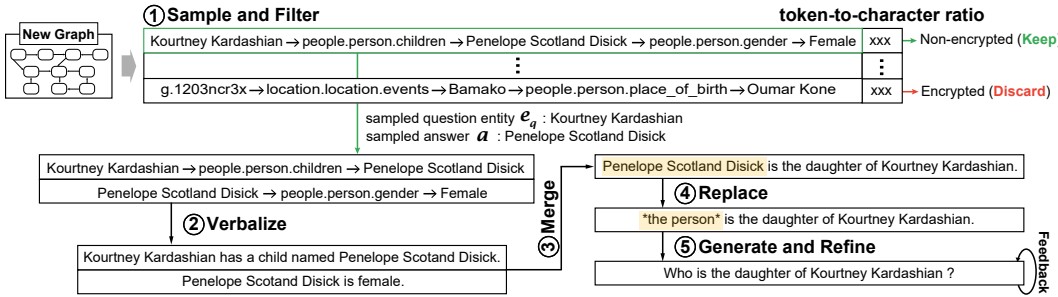

Figure 2: Adaptation Pipeline: stepwise generation of annotations. (Details in Sec. 4.3)

**Step 1. Sample and Filter Reasoning Paths**

Reasoning paths are sampled randomly from the knowledge graph. Paths containing encrypted entities are filtered out, identified by a high token-to-character ratio after tokenization, which indicates non-natural character sequences.

**Step 2. Verbalize Filtered Triples**

Filtered triples are converted into natural language sentences, allowing LLMs to process them more effectively as text rather than structured data.

**Step 3. Construct a Merged Sentence**

The verbalized triples are combined into a single sentence describing the answer. This process helps the model identify relationships between the answer and other information, providing useful expressions for describing them before generating questions, enhancing the final output.

**Step 4. Replace Answer with Placeholder**

The answer in the merged sentence is replaced with a type-indicating placeholder, preventing its direct inclusion.

**Step 5. Generate and Refine Question**

The LLM generates a question from the masked sentence. If the question lacks the question entity—the starting point for reasoning—a refinement step is applied to ensure its inclusion. The generated annotations are then used to optimize the hashing and reasoning modules through supervised loss functions (Equations 3 and 6). Prompts for MobileKGQA and baseline methods, along with detailed annotation examples, are in Appendices K, L, and M.

## 5 EXPERIMENTS

**Datasets.** Following previous works (Mavromatis & Karypis, 2022; Sun et al., 2024; Jiang et al., 2023a; Luo et al., 2024a), we tested our model with two standard KGQA datasets: WebQSP (Yih et al., 2016), CWQ (Talmor & Berant, 2018). More details on the datasets are in Appendix E.

**Baselines.** We compared our method with 8 baselines from 4 categories: 1) IR-based model: ReaRev (Mavromatis & Karypis, 2022), Nutrea (Choi et al., 2023); 2) SP-based model: UnifiedSKG (Xie et al., 2022); 3) LLM-based model: (fine-tuning): RoG (Luo et al., 2024c), GNN-RAG (Mavromatis & Karypis, 2025), SubgraphRAG (Li et al., 2025); 4) LLM-based (ICL) model: StructGPT (Jiang et al., 2023a), ToG (Sun et al., 2024). See Appendices F and D for more details.

**Evaluation Metrics.** Following the prior works (Mavromatis & Karypis, 2025; Li et al., 2025), we measured the performance of our method with Hit and F1-score.[1] Additionally, the hashing module performance was evaluated using mAP, a widely used metric in hashing task (Luo et al., 2023). More details about the metric are in Appendix G.

### 5.1 PERFORMANCE WITHOUT DISTRIBUTION SHIFTS

To assess MobileKGQA in both training and inference, we conducted an analysis against existing KGQA models, focusing on computational resource consumption (see Table 2a) and performance (see Table 2b). One of the most notable observations is that MobileKGQA not only requires the shortest training time among leading lightweight models but also uses only 9% of the parameters

---

[1] Some baselines claim to use Hits@1 for evaluation but actually use Hit, as stated in Li et al. (2025).

and 7.2% of the computational cost (measured under the GTE-large setting). Furthermore, it is the second most optimal model in terms of memory and storage efficiency.

Another key advantage of MobileKGQA is its consistent computational requirements, irrespective of the embedding model's size. Unlike MobileKGQA, the current SOTA lightweight model, GNN-RAG, must repeatedly generate embeddings during training. Consequently, as the embedding model size increases, GNN-RAG's computational cost can far exceed MobileKGQA's by a factor of 41.8 (as measured with GTE-Qwen2-1.5B). In stark contrast, MobileKGQA maintains a constant computational cost regardless of the embedding model, enabling it to leverage large-scale embedding models—which are rapidly advancing for multilingual support and performance enhancement (Hugging Face, 2025)—even in mobile environments.

In terms of performance, MobileKGQA also demonstrates its effectiveness across various embedding models and on-device LLMs (Table 2b). It shows that MobileKGQA achieves performance that is empirically indistinguishable from the current SOTA model, GNN-RAG. Specifically, across all combinations reported, MobileKGQA achieved an average of 0.46 higher Hit and 0.16 lower F1 score compared to GNN-RAG. Even in the worst-case scenario, MobileKGQA recorded only 0.2 lower Hit and 1.2 lower F1 score. These results indicate that the performance difference between MobileKGQA and GNN-RAG is practically negligible. Moreover, this consistency suggests that the hashing module in MobileKGQA can effectively compress embeddings from various embedding models. Compared to ICL-based methods, MobileKGQA demonstrated superior latency and performance across a wide range of LLMs, from mobile (0.5B) to desktop (14B) LLMs.

To more rigorously evaluate the mobile applicability of MobileKGQA, we evaluated it on the NVIDIA Jetson Orin Nano, which provides a resource-constrained environment closely resembling real-world edge-device conditions (Table 3). Under the platform, the 256-bit hash-code–based MobileKGQA could be trained with its optimal hyperparameter configuration, but GNN-RAG could not be trained with the reported optimal configuration due to computational limitations. Accordingly, we tuned GNN-RAG to train within the constraints and selected D1 of WebQSP (see Section 5.2) as a dataset to improve the training stability of GNN-RAG (Details in the Appendix H). Experimental results demonstrate that MobileKGQA can be trained using only 30.6% of the training time and 30.4% of the energy consumption required by GNN-RAG. This efficiency allows an on-device LLM to incorporate new information more rapidly, while also reducing battery drain and enhancing device safety. Moreover, unlike GNN-RAG, MobileKGQA completed training in both 7W and 15W modes without triggering any throttling, underscoring its stability for mobile deployment. Additionally, MobileKGQA exhibits higher GPU utilization and temperature, attributable to the lightweight hash codes that allow faster data loading from the storage and computation. In terms of performance, MobileKGQA further achieved 20.3% improvement over GNN-RAG, showing that simply down-

Table 2: Comparative analysis of training cost and performance of various KGQA models on the WebQSP (P: tuning parameters, Disk: storage, *: LMsr (110M), †: GTE-large (434M), ‡: GTE-Qwen2-1.5B, **red**: best, underline: second-best. **MobileKGQA utilized 256bit hash code and its cost is independent of the embeddings.** Experimental results on CWQ are in Appendix I.1)

(a) Computational cost for training

| Model | P(M) | PFLOPs | T(h) | VRAM(GB) | Disk(GB) |
|---|---|---|---|---|---|
| NuTrea* | 1.2 | 10.32 | 3.3 | 8.6 | **0.51** |
| UnifiedSKG | 2850 | $\geqslant 10^3$ | OOM | $\geqslant 32$ | 1.4 |
| RoG | 6738 | $\geqslant 10^3$ | OOM | $\geqslant 32$ | 2.9 |
| GNN-RAG* | 0.8 | 2.4 | 1.85 | 4.9 | 0.54 |
| GNN-RAG† | 1.1 | 8.3 | 2.10 | 10.4 | 0.54 |
| GNN-RAG‡ | 1.7 | 25.1 | 2.22 | 26.3 | 0.54 |
| SubgraphRAG† | 4.2 | 5.3 | 1.74 | **1.8** | 18.12 |
| MobileKGQA | **0.1** | **0.6** | **1.63** | 3.5 | 0.6 |

(b) Performance on inference

| Base Model | Model | Token | Time(s) | Hit | F1 |
|---|---|---|---|---|---|
| Sentence BERT | ReaRev NuTrea | N/A | 0.08 0.12 | N/A | 70.9 72.7 |
| T5-large | UnifiedSKG | N/A | OOM | N/A | 73.9 |
| Llama 2 (7B) | RoG | 88.5 | OOM | 85.7 | 70.8 |
| Qwen 2 (0.5B, 4bit) | StructGPT | **56.8** | **0.67** | 24.6 | N/A |
| | ToG | 2379.2 | 27.1 | 12.6 | |
| | GNN-RAG* | 68.1 | 0.74 | 68.2 | 43.8 |
| | SubgraphRAG† | 100.9 | 1.18 | 25.6 | 6.4 |
| | MobileKGQA* | 68.6 | 0.72 | **70.6** | **44.2** |
| Gemma 2 (2.6B, 4bit) | StructGPT | 48.6 | 1.02 | 53.8 | N/A |
| | ToG | 2816.3 | 53.7 | 31.6 | |
| | GNN-RAG* | 22.6 | 0.38 | 79.5 | 66.7 |
| | GNN-RAG† | 21.9 | 0.37 | **80.0** | 66.9 |
| | GNN-RAG‡ | **21.6** | **0.36** | 78.5 | 65.6 |
| | SubgraphRAG† | 218.1 | 3.72 | 41.0 | 12.3 |
| | MobileKGQA* | 23.5 | 0.39 | 79.5 | 65.7 |
| | MobileKGQA† | 22.7 | 0.37 | 79.8 | **67.0** |
| | MobileKGQA‡ | **21.6** | **0.36** | 79.8 | **67.0** |
| Llama 3.1 (8B, 4bit) | StructGPT | 84.2 | 2.68 | 53.7 | N/A |
| | ToG | 2426.5 | 65.2 | 51.4 | |
| | GNN-RAG* | 33.8 | 1.19 | 80.1 | **62.0** |
| | SubgraphRAG† | 151.5 | 4.97 | 57.7 | 42.8 |
| | MobileKGQA* | **33.4** | **1.13** | **80.4** | 60.8 |
| Phi 4 (14B, 4bit) | StructGPT | 341.6 | 16.92 | 46.4 | N/A |
| | ToG | 2458.8 | 96.0 | 38.2 | |
| | GNN-RAG* | **49.2** | **2.43** | 82.1 | **62.3** |
| | SubgraphRAG† | 268.6 | 14.1 | 54.5 | 42.1 |
| | MobileKGQA* | 49.7 | 2.46 | **83.0** | 62.2 |

Table 3: Resource usage and KGQA performance of MobileKGQA vs. GNN-RAG under 7W and 15W power modes on the **NVIDIA Jetson platform** (*: Whether the best configuration determined on server can be applied, †: CPU–GPU unified memory, ‡: reasoning-module performance, MobileKGQA utilized 256-bit hash codes. **Red** indicates **MobileKGQA** outperforms, **blue** otherwise.)

| Mode | Model | Best* Config | Training Time (h) | RAM† (GB) | CPU (% / °C) Usage | CPU (% / °C) Temp. | GPU (% / °C) Usage | GPU (% / °C) Temp. | Energy (Wh) | Throttle | RM‡ Hit | RM‡ F1 |
|---|---|---|---|---|---|---|---|---|---|---|---|---|
| 7W | GNN-RAG | impossible | 5.9 | 7.3 | 25.7 | 49.3 | 55.7 | 49.7 | 28.7 | No | 61.7 | 54.3 |
| | MobileKGQA | possible | 2.1 | 5.0 | 23.5 | 48.9 | 70.1 | 50.1 | 11.8 | No | 74.2 | 62.9 |
| | comparison | **Ours better** | **64.4%↓** | **31.5%↓** | **8.6%↓** | **0.8%↓** | **25.9%↑** | **0.8%↑** | **58.9%↓** | Tie | **20.3%↑** | **15.8%↑** |
| 15W | GNN-RAG | impossible | 3.6 | 7.3 | 19.9 | 50.9 | 50.4 | 50.0 | 22.5 | Yes | 61.7 | 54.3 |
| | MobileKGQA | possible | 1.1 | 5.0 | 18.4 | 49.6 | 55.5 | 51.4 | 6.84 | No | 74.2 | 62.9 |
| | comparison | **Ours better** | **69.4%↓** | **31.5%↓** | **7.5%↓** | **2.6%↓** | **10.1%↑** | **2.8%↑** | **69.6%↓** | **Ours better** | **20.3%↑** | **15.8%↑** |

sizing by changing hyperparameters of the model cannot preserve reasoning module performance. These findings highlight the necessity of the proposed hashing strategy for the KGQA system on mobile devices.

## 5.2 PERFORMANCE WITH DISTRIBUTION SHIFTS

To evaluate the proposed adaptation method under distribution shift caused by data accumulation, we designed the following experiment. The dataset was split into three distinct domains, D1, D2, and D3, each further divided into train, validation, and test sets (details in Appendix J). The domain shift was simulated in three stages: the model was first trained on D1 only; then adapted using annotations from D1 and graph data from D1 and D2; and finally adapted again using D1 annotations and graph data from all domains. Table 4 demonstrates that the proposed adaptation method is effective in a domain shift scenario. The model was first trained only on D1 using the train set of D1, and the evaluation results on the test set of D1 are shown in the first column ($S_1$). The first adaptation is then performed on newly accumulated D2 using query-answer pairs generated through annotation generation (see Section 4.3). The results of the adapted model on the test set of D1 and the test set of D2 are shown in the second ($S_1(PA)$) and third ($T_2$) columns, respectively. The results of the above model using both the test set of D1 and D2 are shown in the fourth ($S_{12}$) column. Subsequently, the second adaptation was performed on the newly accumulated D3. After the second adaptation, the performance of model on the test sets of D1 and D2, as well as the test set of D3, are shown in the fifth ($S_{12}(PA)$) and sixth ($T_3$) columns, respectively. Finally, the results obtained from evaluating the model on the test sets of D1, D2, and D3 are presented in the last column (**total**).

Table 4: Hit score of various KGQA models on graph datasets for WebQSP and CWQ datasets across different domains based on Gemma 2 (2B, 4bit) model. (**red**: best results)

| Dataset | WebQSP | | | | | | | CWQ | | | | | | |
|---|---|---|---|---|---|---|---|---|---|---|---|---|---|---|
| Model | $S(D1) \rightarrow T(D2)$ | | | $S(D1+D2) \rightarrow T(D3)$ | | | | $S(D1) \rightarrow T(D2)$ | | | $S(D1+D2) \rightarrow T(D3)$ | | | |
| | $S_1$ | $S_1(PA)$ | $T_2$ | $S_{12}$ | $S_{12}(PA)$ | $T_3$ | total | $S_1$ | $S_1(PA)$ | $T_2$ | $S_{12}$ | $S_{12}(PA)$ | $T_3$ | total |
| ToG | 31.3 | | 30.9 | 31.1 | | **32.1** | 31.6 | 29.3 | | 32.1 | 30.9 | | 29.9 | 30.6 |
| ReaRev | 76.2 | | 30.0 | 58.1 | | 20.4 | 41.4 | 41.0 | | 15.7 | 26.4 | | 14.7 | 22.4 |
| NuTrea | 69.6 | N/A | 25.4 | 52.3 | N/A | 12.5 | 34.7 | 37.3 | N/A | 9.62 | 21.3 | N/A | 23.5 | 22.1 |
| GNN-RAG | **79.2** | | 36.4 | 62.5 | | 27.4 | 47.0 | **57.9** | | 31.6 | 42.7 | | 32.1 | 39.1 |
| SubgraphRAG | 22.9 | | 15.6 | 20.0 | | 14.3 | 17.5 | 30.1 | | 10.3 | 18.7 | | 5.1 | 14.0 |
| MobileKGQA | 79.1 | 78.9 | **43.2** | **64.9** | 63.4 | 30.4 | **48.8** | 57.8 | 60.2 | **34.0** | **45.1** | 45.6 | **32.5** | **41.0** |

Examining the experimental results, we find that MobileKGQA outperforms the baselines, highlighting the effectiveness of our annotation generation method. Notably, in the CWQ dataset, performance on the original domain improved after adaptation (i.e., $S_1$ vs. $S_1(PA)$, $S_{12}$ vs. $S_{12}(PA)$), suggesting that annotations generated for the new domain successfully transferred knowledge. These results confirm the effectiveness of our adaptation strategy. Additionally, comparison with few-shot learning methods and results under data deletion scenarios are in appendices I.2 and I.3. Moreover, the adaptation cost measured on the NVIDIA Jetson platform is in Appendix I.4.

# 6 ANALYSIS

## 6.1 IMPACT OF HASHING DIMENSION ON PERFORMANCE AND COMPUTATIONAL COST

Table 5: Changes in Performance and training costs based on hashing dimensions (Using Gemma 2 (2B, 4bit) and LMsr models, RM: Reasoning Module only, Disk: storage for embeddings)

| Metric | WebQSP | | | | CWQ | | | |
|---|---|---|---|---|---|---|---|---|
| | FP32 768 | 256bit | 128bit | 64bit | FP32 768 | 256bit | 128bit | 64bit |
| mAP | 1 | 0.907 | 0.877 | 0.827 | 1 | 0.887 | 0.845 | 0.774 |
| Hit | 79.5 | 79.5 | 79.1 | **78.4 (1.3%↓)** | 54.4 | 54.4 | 53.8 | **53.5 (2.9%↓)** |
| Hit (RM) | 77.6 | 77.5 | 76.6 | **75.5 (2.7%↓)** | 41.4 | 41.3 | 40.6 | **40.3 (2.7%↓)** |
| Disk(GB) | 6.2 | 0.065 | 0.032 | 0.016 | 89.3 | 0.93 | 0.46 | 0.23 |
| PFLOPs | 1.46 | 0.36 | 0.29 | 0.26 | 8.50 | 2.63 | 2.32 | 2.14 |
| Params(M) | 0.239 | 0.136 | 0.130 | 0.127 | 0.291 | 0.189 | 0.183 | 0.179 |

We conducted an analysis to demonstrate that the hashing module effectively preserves semantics while reducing the model's computational costs. Table 5 shows the performance and computational cost for different hashing dimensions in MobileKGQA. Specifically, FP32 768 represents the original 768-dimensional embedding generated by the LMsr model (Zhang et al., 2022). Consistent with previous experiments, MobileKGQA and its retrieval module show minimal degradation—less than 2.9%—at 64 bits, while maintaining near-identical performance up to 256 bits on both datasets. This demonstrates that our hashing method effectively maintains performance while reducing computational resources. Furthermore, on the WebQSP dataset, the 64-bit hash code achieves a strong mAP of 0.75, indicating high-quality hashing. Compared to the original embedding, hashing reduces storage by 99.75%, and reduces the Reasoning Module's parameters and PFLOPs by 46.9% and 82.2%, respectively. Additional experiments on hashing module training costs and annotation generation costs are in Appendix I.5 and I.6.

## 6.2 QUALITY ANALYSIS ON GENERATED QUERIES

Table 6: Quality analysis on the generated annotations using various LLMs and WebQSP dataset. (**red**: best, blue: second-best. Full table is provided in Appendix I.7.)

| Method | Gemma2 (2B, 4bit) | | | | Phi 4 (14B, 4bit) | | | |
|---|---|---|---|---|---|---|---|---|
| | ROUGE-L(%) | BERTScore(%) | Tokens | Time(s) | ROUGE-L(%) | BERTScore(%) | Tokens | Time(s) |
| RLM | 17.0 | 36.4 | **25401** | **6.8** | 12.1 | 18.8 | **54045** | **16.9** |
| CoT | 32.3 | 37.6 | 328687 | 53.2 | 26.0 | 32.5 | 424289 | 118.2 |
| Ours | **42.8** | **48.7** | 97855 | 25.5 | **41.9** | **48.4** | 89471 | 30.2 |

To evaluate the quality of annotations generated by MobileKGQA, we compared the method with two existing approaches: ReasoningLM (RLM) (Jiang et al., 2023b) and Chain of Thought (CoT) (Wei et al., 2022) (prompt in Appendix L). Specifically, we used human-generated questions in WebQSP as references and compared them with questions generated by various methods. As shown in Table 6, MobileKGQA outperformed all baselines in both ROUGE-L and BERTScore, clearly demonstrating the effectiveness of our method. Moreover, MobileKGQA required only 21% of the tokens and 26% of the time compared to CoT (measured using Phi-4), demonstrating its efficiency across various models—from mobile-scale (2B) to desktop-scale (14B).

## 6.3 ABLATION STUDY

To evaluate the effectiveness of our proposed annotation generation method, we compared its performance against RLM and CoT. Table 7 presents the adaptation performance when adapting from D1 to D2 in WebQSP. The results show that MobileKGQA consistently achieved the highest performance across all domains, clearly demonstrating the effectiveness of our method. Experiments were conducted on the WebQSP dataset using the Gemma 2 (2B, 4-bit).

Table 7: Ablation study on the annotation generation method

| Method | Hit | | | |
|---|---|---|---|---|
| | S | S(PA) | T | total |
| RLM | 79.1 | 78.1 | 37.8 | 62.3 |
| CoT | 79.1 | 73.6 | 42.4 | 61.4 |
| Ours | 79.1 | **78.9** | **43.2** | **64.9** |

## 7 CONCLUSION

In this paper, we present MobileKGQA, an on-device knowledge-graph question answering system designed for on-device training. To reduce resource demands, a hashing module converts high-dimensional embeddings into hash codes, while a stepwise reasoning method enables annotation generation with minimal tokens. Mobile applicability is demonstrated on the NVIDIA Jetson Orin Nano, where the system delivers 20.3% higher performance and uses only 30.4% of the energy of the state-of-the-art model. Moreover, using just 9% of the parameters and 7.2% of the computation of the SOTA baseline, MobileKGQA attains comparable performance and outperforms baselines in evolving databases. This work establishes a foundation for practical, privacy-preserving KGQA systems for dynamic mobile environments.

## 8 ACKNOWLEDGEMENTS

This research was supported by the National Research Foundation of Korea (NRF) funded by the Korean government (MSIT) under Grant Nos. 2022R1A3B1077720, 2022R1A5A7083908, and RS-2022-NR067569 (Pioneer Research Center Program). This work was also supported by the BK21 FOUR Program (Education and Research Program for Future ICT Pioneers) at Seoul National University in 2025. In addition, this research was supported by the Institute of Information & Communications Technology Planning & Evaluation (IITP) grant funded by MSIT (RS-2021-II211343, Artificial Intelligence Graduate School Program at Seoul National University). Furthermore, this work was conducted as part of the Sovereign AI Foundation Model Project (Data Track) organized by MSIT and supported by the National Information Society Agency (NIA), Republic of Korea (Grant No. 2026-AIData-WII01). Moreover, this work was supported by HUINNO AIM Company through HA-Rnd2325-predictClinicalDeterioration.

## 9 REPRODUCIBILITY STATEMENT

To enhance the reproducibility of this study, we provide detailed proofs, dataset descriptions, and comprehensive implementation details with hyperparameters, in Appendices C, D, E, respectively.

## 10 ETHICS STATEMENT

This work relies solely on publicly available datasets for evaluation. The proposed on-device framework aims to enhance privacy by enabling local processing of user data, thereby reducing reliance on centralized servers. However, in practical deployments, appropriate safeguards should be implemented to ensure the secure handling of potentially sensitive information. As the system incorporates LLM-based reasoning and automated annotation generation, it may produce inaccurate or misleading outputs; therefore, it is not intended for use in high-stakes or safety-critical applications without appropriate human oversight. We also acknowledge that personalization systems may introduce unintended biases depending on the data, and future work will focus on mitigating such risks.

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

# Appendix for *MobileKGQA: On-Device KGQA System on Dynamic Mobile Environments*

## A  DETAILS ABOUT THE USE OF LARGE LANGUAGE MODELS

In this study, we utilized large language models (LLMs) for manuscript preparation, specifically for refining sentences and checking grammar and spelling. The authors carefully reviewed and edited the content generated by the LLMs to ensure accuracy and consistency. Additionally, the LLMs did not influence the research ideas or interpretation of results.

## B  ADDITIONAL RELATED WORK

### B.1  IR-BASED METHODS

- **GRAFT-Net** (Sun et al., 2018) leverages graph representation learning to answer questions by extracting information from a hybrid subgraph that integrates both knowledge base and entity-linked text resources.

- **PullNet** (Sun et al., 2019) is an iterative framework that constructs a question-specific subgraph through a graph convolutional network-based retrieval process, enabling multi-hop reasoning across both knowledge base and corpus resources.

- **NSM** (He et al., 2021) utilizes a teacher-student framework where a bidirectional (forward/backward) reasoning teacher network effectively guides the student network's intermediate step learning in knowledge base question answering.

- **EmbedKGQA** (Saxena et al., 2020) leverages knowledge graph embedding techniques to address the challenge of incomplete knowledge graphs in multi-hop question answering, eliminating the need for external text resources or neighborhood constraints.

- **TransferNet** (Shi et al., 2021) performs multi-hop reasoning by sequentially transferring entity scores through activated relations, supporting both knowledge graph labels and text relations while maintaining interpretability through transparent intermediate steps.

- **ReaRev** (Mavromatis & Karypis, 2022) enhances knowledge graph question answering through adaptive instruction refinement and graph neural network-based breadth-first search execution, enabling flexible reasoning that adapts to available knowledge graph information.

- **NuTrea** (Choi et al., 2023) is a tree search-based graph neural network that enhances multi-hop knowledge graph question answering by incorporating both forward-looking message passing and global context through a novel node embedding scheme.

### B.2  SP-BASED METHODS

- **QGG** (Lan & Jiang, 2020) enhances knowledge base question answering by integrating constraint processing early in the query graph generation process, utilizing a staged approach that efficiently handles both multi-hop relations and constrained queries simultaneously.

- **ReTraCk** (Chen et al., 2021) is a modular framework that combines a retriever for knowledge base item selection, a transducer for logical form generation, and a checker for verification, enabling efficient question-answering over large-scale knowledge bases.

- **UnifiedSKG** (Xie et al., 2022) proposes a unified text-to-text framework that standardizes diverse structured knowledge grounding tasks for more systematic and compatible research across different domains.

- **TIARA** (Shu et al., 2022) enhances knowledge base question answering by combining pretrained language models with multi-grained retrieval and constrained decoding, enabling improved semantic understanding and logical form generation.

- **DecAF** (Yu et al., 2023) presents a hybrid framework that combines logical form generation and direct answer prediction for knowledge base question answering while simplifying the process through free-text retrieval.
- **UniKGQA** (Jiang et al., 2023c) presents a unified approach that combines Knowledge Graph retrieval and reasoning processes using a pre-trained language model and matching information propagation for multi-hop question answering.

### B.3 LLM-BASED METHODS

- **StructGPT** (Jiang et al., 2023a) presents a framework called StructGPT that enhances language models' reasoning on structured data through an iterative two-step approach of collecting evidence via specialized interfaces before performing targeted reasoning to gradually reach the correct answer.
- **Pangu** (Gu et al., 2023) presents a grounded language understanding framework that leverages language models' discriminative abilities rather than generative capabilities by combining a symbolic agent for environment exploration with a neural language model that evaluates plan plausibility.
- **KB-BINDER** (Li et al., 2023a) enables few-shot knowledge base question answering by using large language models to generate logical forms based on demonstrations, then grounding them to executable queries through BM25 score matching.
- **ToG** (Sun et al., 2024) presents an interactive agent-based approach that enables large language models to perform verifiable reasoning by iteratively exploring and leveraging knowledge graphs through beam search to find optimal reasoning paths and generate reliable answers.
- **FlexKBQA** (Li et al., 2024) presents a few-shot framework called FlexKBQA that leverages large language models to translate sampled knowledge base programs into natural language questions, then uses execution-guided self-training with unlabeled user queries to train a lightweight specialized model.
- **SymKGQA** (Agarwal et al., 2024) is a few-shot KGQA framework that uses in-context learning to create step-by-step symbolic logical forms. It stands out from previous methods by not requiring the LLM to have prior knowledge of KoPL. To connect these logical forms to the knowledge graph, SymKGQA includes a retrieval-augmented module called QUACK, which provides grounding based on the specific question.

## C PROOF OF THEOREM 1

**Assumption 1** (Independence Assumption). *The norm of LLM embeddings $r_z$ is independent of both the orientation of embeddings $\boldsymbol{\theta_z}$ and the orientation of hash codes $\boldsymbol{\theta_h}$, i.e.,*

$$P(r_z, \boldsymbol{\theta_z}) = P(r_z)P(\boldsymbol{\theta_z})$$
$$P(r_z, \boldsymbol{\theta_h}) = P(r_z)P(\boldsymbol{\theta_h})$$

**Theorem 1** (Conditions for Mutual Information Maximization). *Suppose Assumptions 1 and 2 hold, and both $\phi$ and $\psi$ preserve the cosine similarity between any pair of embeddings in Z. Then, the mutual information $I(z, h)$ is maximized.*

*Proof.* By definition, the mutual information between $z$ and $h$ is given by:

$$I(z, h) = H(z) - H(z \mid h). \tag{7}$$

Expressing $z$ in polar coordinates,

$$I(z, h) = H(r_z, \theta_z) - H(r_z, \theta_z \mid h). \tag{8}$$

Applying the first condition of Assumption 1,

$$H(r_z, \theta_z) = H(r_z) + H(\theta_z), \tag{9}$$

and similarly,

$$H(r_z, \theta_z \mid h) = H(r_z \mid h) + H(\theta_z \mid h). \tag{10}$$

Thus, we can rewrite mutual information as:

$$I(z, h) = H(r_z) + H(\theta_z) - H(r_z \mid h) - H(\theta_z \mid h). \tag{11}$$

Using the definition of mutual information,

$$I(z, h) = I(r_z, h) + I(\theta_z, h). \tag{12}$$

Now, consider the hash code $h \in \{-1, 1\}^d$. Since each element of $h$ takes values in $\{-1, 1\}$, $r_h$ is a constant. Consequently,

$$I(z, h) = I(r_z, \theta_h) + I(\theta_z, \theta_h). \tag{13}$$

Applying the second condition of Assumption 1,

$$I(z, h) = I(\theta_z, \theta_h). \tag{14}$$

By the definition of mutual information,

$$I(z, h) = H(\theta_z) - H(\theta_z \mid \theta_h) \leqslant H(\theta_z). \tag{15}$$

Since both $\phi$ and $\psi$ preserve cosine similarity, the $\theta_h$ determines $\theta_z$, ensuring

$$H(\theta_z \mid \theta_h) = 0. \tag{16}$$

Thus, mutual information is maximized:

$$I(z, h) = H(\theta_z). \tag{17}$$

$\square$

# D  IMPLEMENTATION DETAILS

## D.1  HYPERPARAMETER SETTINGS

Table 8: Hyperparameters for Hashing Module

| Embedding | Dataset | Bit | lr | $\alpha$ |
|---|---|---|---|---|
| LMsr | WebQSP | 64bit | $1.0 \times 10^{-5}$ | 0.01 |
| | | 128bit | $1.0 \times 10^{-5}$ | 0.01 |
| | | 256bit | $1.0 \times 10^{-5}$ | 0.01 |
| | CWQ | 64bit | $5.0 \times 10^{-5}$ | 0.01 |
| | | 128bit | $5.0 \times 10^{-5}$ | 0.01 |
| | | 256bit | $1.0 \times 10^{-5}$ | 0.01 |
| GTE-large | WebQSP | 64bit | $5.0 \times 10^{-5}$ | 0.01 |
| | | 128bit | $5.0 \times 10^{-5}$ | 0.01 |
| | | 256bit | $5.0 \times 10^{-5}$ | 0.01 |
| GTE-Qwen2-1.5B | WebQSP | 64bit | $1.0 \times 10^{-4}$ | 0.01 |
| | | 128bit | $5.0 \times 10^{-5}$ | 0.01 |
| | | 256bit | $3.0 \times 10^{-5}$ | 0.01 |

Table 8 presents the learning rate (lr) and balancing parameter ($\alpha$) settings used for hashing module across various LLMs on the WebQSP and CWQ datasets. Table 9 provides the hyperparameter settings for the reasoning module on the WebQSP and CWQ datasets, including the number of iterations (# iter.), instructions (# ins.), GNN layers (# gnn), and embedding dimensions (dim). In this study, all reported experiments were conducted three times, and the average results are presented. The random seeds 0, 1, and 2 were used for the repetitions. All experiments were conducted on a NVIDIA Tesla V100 GPU.

Table 9: Hyperparameters for Reasoning Module

| Embedding | Bit | lr. | # iter. | # ins. | # gnn | dim |
|---|---|---|---|---|---|---|
| **WebQSP** | | | | | | |
| LMsr | 64bit | 0.0009 | 3 | 2 | 2 | 50 |
| LMsr | 128bit | 0.0009 | 3 | 2 | 2 | 50 |
| LMsr | 256bit | 0.0007 | 3 | 2 | 2 | 50 |
| GTE-large | 256bit | 0.0004 | 2 | 3 | 3 | 50 |
| GTE-Qwen2-1.5B | 256bit | 0.0005 | 2 | 3 | 3 | 50 |
| **CWQ** | | | | | | |
| LMsr | 64bit | 0.0007 | 2 | 3 | 3 | 50 |
| LMsr | 128bit | 0.0007 | 3 | 2 | 3 | 50 |
| LMsr | 256bit | 0.0005 | 2 | 3 | 3 | 50 |

## D.2 UTILIZED QUANTIZED LLMS IN THIS STUDY

In this study, we evaluated the performance of MobileKGQA in comparison with baseline models across various large language models. Our experiments revealed that among the open-source quantized large language models, those provided by Ollama exhibited the best text generation performance while enabling optimized computations across diverse local machines. Consequently, we employed Ollama's models for the text generation stage in both the baseline models and MobileKGQA's LLM. Specific Ollama models used in this study can be found at the following link.

- Qwen2-0.5B (`https://ollama.com/library/qwen2:0.5b`)
- Gemma2-2B (`https://ollama.com/library/gemma2:2b`)
- Llama3.1-8B (`https://ollama.com/library/llama3.1`)
- Phi4-14B (`https://ollama.com/library/phi4`)

## E DATASET DETAILS

Table 10: Statistics of datasets. Each row represents the number of Questions, Questions in train/validation/test set, total entities, entities per question, total relations, relations per question, total triples, triples per question, followed by the knowledge base that the datasets originated from.

| Attribute | WebQSP | CWQ |
|---|---|---|
| Total Questions | 4,700 | 34,689 |
| Train | 2826 | 27,639 |
| Validation | 246 | 3,519 |
| Test | 1,628 | 3,531 |
| Total Entities | 1,298,306 | 2,259,510 |
| Entities per Question | 2,461 | 11,422 |
| Total Relations | 6,094 | 6,649 |
| Relations per Question | 628 | 845 |
| Total Triples | 19,986,134 | 138,785,703 |
| Triples per Question | 4,252 | 4,001 |
| Knowledge Base | Freebase | Freebase |

In this study, we evaluate our approach using two prominent KGQA datasets: WebQSP (Yih et al., 2016) and CWQ (Talmor & Berant, 2018), both constructed from the Freebase knowledge graph. The knowledge graphs contain 1.29 million and 2.25 million entities, respectively. To ensure a fair comparison with existing research, we maintain the established train-dev-test splits from previous work (Mavromatis & Karypis, 2022; Yu et al., 2023; Sun et al., 2024). Specifically, we utilize the

preprocessed version of the datasets from Luo et al. (2024c), which is available through the Hugging Face platform (WebQSP[2], CWQ[3]). The detailed statistics of these datasets are presented in Table 10. WebQSP exhibits varying levels of reasoning complexity: 65.49% can be answered using a single fact, while 34.51% of questions require aggregation across two facts. The maximum reasoning depth necessary is limited to two hops within the graph. On the other hand, CWQ surpasses WebQSP in complexity and requires more sophisticated reasoning patterns. The questions are categorized as 45% composition, 45% conjunction, 5% comparative, and 5% superlative. In terms of reasoning depth, 40.91% of questions are answerable with a single fact, 38.34% require aggregation across two facts, and 20.75% involve reasoning over more than three facts. The maximum reasoning depth extends to four hops within the graph.

## F  BASELINE SELECTION CRITERIA AND DETAILS

To rigorously evaluate our model, we selected baselines based on the following criteria: in scenarios without distribution shift, we compared our model to high-performance, large-parameter fine-tuned models (Xie et al., 2022; Luo et al., 2024c). In scenarios with distribution shifts, we chose the top-performing models with lightweight design, either capable of fine-tuning in mobile environments (Mavromatis & Karypis, 2025; Li et al., 2025) or not requiring fine-tuning at all (Jiang et al., 2023a; Sun et al., 2024). Moreover, to reflect the fast-growing on-device LLM field, we selected four newly released on-device LLM models: Qwen2-0.5B, Gemma2-2B, Llama 3.1-8B, and Phi4-14B. Gemma2-2B is an on-device LLM that can operate on mobile devices without restriction. Additionally, to account for advancements in on-device LLM inference speed, we utilized Qwen2-0.5B, and to reflect improvements in on-device LLM performance, we employed Llama 3.1-8B and Phi4-14B, which cannot operate on mobile devices without restrictions yet. As embedding models, we used the LMsr (Zhang et al., 2022) and GTE-large (Zhang et al., 2024), which were employed in GNN-RAG and SubgraphRAG (Mavromatis & Karypis, 2025; Li et al., 2025). Additionally, to account for the rapidly increasing scale of embedding models, we utilized the GTE-Qwen2:1.5B model (Li et al., 2023b), which is the current SOTA embedding model in MTEB dataset (Hugging Face, 2025).

## G  EXPLANATION OF THE METRICS

To evaluate the question-answering performance of MobileKGQA, we utilize two widely used metrics in previous research (Li et al., 2025): F1 score and Hit. Moreover, to evaluate the performance of the hashing module, mean average precision (mAP) is utilized.

**F1 Score.** The F1 score is the harmonic mean of precision and recall, providing a balanced measure of a model's accuracy when both false positives and false negatives are critical. It is defined as:

$$\text{F1} = 2 \times \frac{\text{Precision} \times \text{Recall}}{\text{Precision} + \text{Recall}} \tag{18}$$

**Hit.** Hit measures the proportion of queries for which at least one correct answer appears in the model's output. It is defined as:

$$\text{Hit} = \frac{1}{Q} \sum_{i=1}^{Q} 1[\hat{A}_i \cap A_i \neq \varnothing] \tag{19}$$

where $Q$ is the number of queries, $\hat{A}_i$ is the set of predicted answers for the $i$-th query, $A_i$ is the set of ground truth answers, and $1[\cdot]$ is an indicator function that returns 1 if the condition is met and 0 otherwise.

**Mean Average Precision (mAP)** is a widely used metric for evaluating information retrieval systems, as it considers both precision and the ranking of relevant items. In our evaluation, mAP measures how well the hashing module preserves similarity between LLM representations. mAP is based

---

[2]`https://huggingface.co/datasets/rmanluo/RoG-webqsp`
[3]`https://huggingface.co/datasets/rmanluo/RoG-cwq`

on Precision@$k$, which quantifies the proportion of relevant items among the top-$k$ retrieved results. Given a query $q$, let $\mathcal{R}_k$ denote the set of top-$k$ retrieved items and $\mathcal{G}$ be the set of ground-truth relevant items. Then, Precision@$k$ is defined as:

$$\text{Precision@}k = \frac{|\mathcal{R}_k \cap \mathcal{G}|}{k} \tag{20}$$

Since Precision@$k$ only measures performance at a specific cutoff, the relevance indicator function $\text{rel}(k)$ is utilized to capture whether a retrieved item contributes to precision:

$$\text{rel}(k) = \begin{cases} 1, & \text{if the } k\text{-th retrieved item is relevant} \\ 0, & \text{otherwise} \end{cases} \tag{21}$$

Using these definitions, we compute Average Precision (AP), which measures the average of precision values at positions where relevant items appear. It is given by:

$$\text{AP} = \frac{1}{|\mathcal{G}|} \sum_{k=1}^{N} \text{Precision@}k \cdot \text{rel}(k) \tag{22}$$

where $N$ is the total number of retrieved items. Finally, we define Mean Average Precision (mAP) as the mean of AP values across multiple queries. Given a set of queries $\{q_1, q_2, \ldots, q_Q\}$, mAP is expressed as:

$$\text{mAP} = \frac{1}{Q} \sum_{i=1}^{Q} \text{AP}_i \tag{23}$$

where $\text{AP}_i$ is the average precision for query $q_i$.

To compute mAP in our experiments, we randomly sampled 100 queries from the test dataset and repeated 30 times. For each sampled set, we computed the mAP score and averaged the results to obtain a final evaluation metric. For defining relevant items, we used the five most similar representations among the entire set based on cosine similarity. These top-5 nearest neighbors were considered to belong to the same class for each query.

## H   DETAILS ABOUT THE EXPERIMENTS ON THE NVIDIA JETSON ORIN NANO

Table 11: Key specifications of the NVIDIA Jetson Orin Nano (8 GB)

| Component | Specification |
|---|---|
| GPU | 1024 core NVIDIA Ampere architecture with 32 Tensor Cores |
| CPU | 6-core ARM Cortex-A78AE v8.2 64-bit CPU 1.5 MB L2 + 4 MB L3 |
| Memory | 8 GB 128-bit LPDDR5 @ 68 GB/s |
| Power Modes | Configurable 7 W / 15 W |

The NVIDIA Jetson Orin Nano 8 GB was employed as the on-device AI platform for our study. We chose it as a proxy for real-world mobile and edge devices due to its limited electrical power, small thermal envelope, and compact physical dimensions. Its configurable 7W and 15W power modes enabled experiments under two representative operating conditions. The 7W mode approximates the thermal and power constraints of smartphones or small hand-held robots, where continuous operation is critical. In contrast, the 15W mode serves as a proxy for larger edge platforms such as drones, service robots, or industrial IoT gateways that have slightly higher power and thermal budgets but remain well below desktop-class levels. By combining these tightly controlled power settings with a full GPU–CPU system-on-module, the board effectively reproduces the resource limitations (low power, restricted cooling, and confined form factor) of real-world mobile environments.

To enable a fair comparison between GNN-RAG and MobileKGQA on the Jetson platform, we carefully reproduced the experimental conditions reported in the original GNN-RAG paper. We employed LMsr as the embedding model, since it achieved the best performance in their experiments, and used the authors' official evaluation code without modification. However, the configuration described by the GNN-RAG authors requires substantially more memory than the Jetson platform can provide, making training infeasible. To overcome this limitation, we modified their dataloader to reduce memory and re-tuned the model to enable training within the limited memory constraints. Moreover, the memory demand for relation embeddings in GNN-RAG scales with the size of the dataset. Therefore, we trained the model on D1, a subset of the WebQSP dataset, to keep the overall memory requirements manageable. After these adjustments, we successfully configured the model to operate within the Jetson's 7.6 GB memory limit, occupying 7.3 GB, as summarized in Table 12.

Table 12: Hyperparameter Settings for GNN-RAG on the NVIDIA Jetson platform

| Embedding | batch size | lr. | # iter. | # ins. | # gnn | dim |
|-----------|-----------|-----|---------|--------|-------|-----|
| LMsr | 1 | 0.0005 | 3 | 1 | 2 | 10 |

# I  ADDITIONAL EXPERIMENTS

## I.1  PERFORMANCE OF MOBILEKGQA ON CWQ AND METAQA DATASET WITHOUT DISTRIBUTION SHIFT.

Table 13: Performance comparison of various KGQA models on the **CWQ** dataset. (**red**: best, blue: second-best, MobileKGQA utilized 256bit hash codes and LMsr embedding model.)

| Base Model | Model | Token | Time(sec.) | Hit | F1 |
|-----------|-------|-------|-----------|-----|-----|
| Sentence BERT | ReaRev
NuTrea | N/A | 0.08
0.12 | N/A | 41.4
49.5 |
| T5 (3B) | UnifiedSKG | N/A | OOM | N/A | 73.3 |
| Llama 2 (7B) | RoG | 89.6 | OOM | 62.6 | 56.2 |
| Qwen 2
(0.5B, 4bit) | ToG | 3010.1 | 36.1 | 15.2 | N/A |
| | GNN-RAG | **83.3** | 0.98 | 47.0 | 29.5 |
| | SubgraphRAG | 86.0 | 0.97 | 17.8 | 6.4 |
| | MobileKGQA | 85.1 | **0.93** | **47.5** | **30.1** |
| Gemma 2
(2B, 4bit) | ToG | 2534.6 | 53.2 | 31.9 | N/A |
| | GNN-RAG | **16.9** | 0.33 | **54.4** | 40.7 |
| | SubgraphRAG | 186.9 | 3.49 | 32.7 | 14.0 |
| | MobileKGQA | 17.0 | **0.32** | **54.4** | **40.8** |
| Llama 3.1
(8B, 4bit) | ToG | 3540.4 | 83.6 | 39.4 | N/A |
| | GNN-RAG | 44.0 | 1.17 | 57.6 | **37.3** |
| | SubgraphRAG | 269.0 | 6.33 | 40.2 | 30.7 |
| | MobileKGQA | **43.7** | **1.01** | **57.8** | 37.1 |
| Phi 4
(14B, 4bit) | ToG | 3402.3 | 166.5 | 18.4 | N/A |
| | GNN-RAG | **88.9** | **4.41** | **64.2** | **39.0** |
| | SubgraphRAG | 404.5 | 21.1 | 39.3 | 33.2 |
| | MobileKGQA | 90.4 | 4.47 | 63.8 | 38.8 |

In the main paper, we evaluated the computational cost and performance of various KGQA models on the WebQSP dataset. In this section, we additionally report experimental results on the CWQ dataset. (See Table 13). Consistent with the findings on WebQSP, MobileKGQA achieves performance that is nearly identical to the state-of-the-art lightweight model, GNN-RAG, in the CWQ dataset. The average performance gap between the two models is only 0.075 in both Hit score and F1 score. Furthermore, MobileKGQA requires the least amount of resources in terms of latency and token generation, demonstrating its practical efficiency.

We further evaluate MobileKGQA on the MetaQA benchmark, which comprises multi-hop question answering tasks, and compare its performance against the GNN-RAG baseline. As shown in Table 14, MobileKGQA achieves comparable results across 1-hop, 2-hop, and 3-hop settings, closely

matching the baseline scores. These findings indicate that the proposed hashing-based architecture effectively preserves the semantic information necessary for multi-hop reasoning and maintains stable retrieval quality when extended to additional knowledge graph benchmarks.

Table 14: F1 Score Comparison on MetaQA Datasets

| Model | MetaQA 1-hop | MetaQA 2-hop | MetaQA 3-hop |
|---|---|---|---|
| GNN-RAG | $95.9 \pm 0.53$ | $96.7 \pm 0.78$ | $88.7 \pm 0.86$ |
| MobileKGQA | $95.8 \pm 0.68$ | $96.7 \pm 0.63$ | $88.6 \pm 0.92$ |

## I.2 PERFORMANCE COMPARISON WITH FEW-SHOT LEARNING METHOD

Table 15: Comparison of MobileKGQA and Few-Shot Learning Methods on WebQSP (MobileKGQA utilized 256bit hash codes; Hit score reported. *: p-value $\leqslant 0.0001$ )

| Model | On-device LLM | |
|---|---|---|
| | Gemma 2 (2B, 4bit) | Phi-4 (14B, 4bit) |
| SymKGQA | $10.6 \pm 0.4$ | $12.3 \pm 0.2$ |
| MobileKGQA | $\mathbf{37.3 \pm 0.3}$ * | $\mathbf{39.2 \pm 0.4}$ * |

To further analyze the adaptability of MobileKGQA to its shifted distribution, we conducted a comparison with SOTA few-shot learning methods based on cloud-based LLMs: symKGQA (Agarwal et al., 2024). Since obtaining a large number (e.g., over 100) of ground truth annotations is often impractical in real-world scenarios, we evaluated all models under a zero-shot setting to reflect practical constraints. Additionally, the experiments were conducted in a setting where the cloud-based LLM was replaced with an on-device LLM.

Experimental results show that MobileKGQA consistently outperforms symKGQA on both on-device LLMs. (See Table 15) This is because conventional few-shot learning methods rely solely on few-shot prompting to translate between natural language and logical forms (e.g., SPARQL, KoPL), which require a lengthy reasoning process or high-capacity cloud-based LLMs. As a result, these methods often fail to execute the translation process effectively in resource-constrained mobile environments, producing irrelevant questions or nonsensical logical forms. In contrast, MobileKGQA decomposes the complex translation process into smaller substeps that are more manageable for on-device LLMs, enabling strong performance even in resource-constrained settings.

## I.3 CONSIDERATION ON DISTRIBUTION SHIFT CAUSED BY **DATA DELETION**

Table 16: Hit scores of GNN-RAG and MobileKGQA on the WebQSP dataset under varying data deletion ratios. (MobileKGQA used 256-bit hash codes. *: p $\leqslant 0.01$, **: p $\leqslant 0.001$.)

| Deletion Ratio | GNN-RAG | MobileKGQA |
|---|---|---|
| 20% | $84.2 \pm 0.2$ | $\mathbf{87.2 \pm 0.4}$ * |
| 40% | $77.2 \pm 0.6$ | $\mathbf{84.9 \pm 0.8}$ ** |
| 60% | $70.0 \pm 0.2$ | $\mathbf{83.3 \pm 0.6}$ ** |
| 80% | $56.7 \pm 0.3$ | $\mathbf{71.4 \pm 0.3}$ ** |

Data deletion is common on mobile devices and can induce distribution shifts that degrade the performance of KGQA systems. Nevertheless, valid annotations for the updated database can still be generated using rule-based methods. Consequently, when data deletion occurs, MobileKGQA can efficiently adapt through on-device supervised training with only minimal computational overhead. In contrast, the current state-of-the-art model, GNN-RAG, must either undergo resource-intensive retraining (see Table 2) or accept a noticeable decline in performance.

Table 16 presents experimental results supporting this claim, showing the performance of models trained on the original WebQSP dataset and evaluated on a version with randomly deleted triples. As

shown in the table, MobileKGQA consistently outperforms GNN-RAG across all deletion scenarios, and even when 80% of the edges are removed, it exhibits only an 18.1% performance drop. This demonstrates that MobileKGQA can effectively adapt even under data deletion conditions.

## I.4 COST OF MODEL ADAPTATION ON THE **NVIDIA JETSON ORIN NANO**

Table 17: Adaptation cost of MobileKGQA on **NVIDIA Jetson Orin Nano** (based on Gemma 2 (2B, 4bit) model, WebQSP dataset)

| Mode | Condition | Energy (Wh) | Time (h) | CPU | | GPU | | RAM (GB) | Throttle |
|------|-----------|-------------|----------|-----------|------------|-----------|------------|----------|----------|
| | | | | Usage (%) | Temp (°C) | Usage (%) | Temp (°C) | | |
| 7 W | S (D1)→T (D2) | 14.2 | 2.5 | 24.1 | 48.7 | 69.8 | 50.3 | 5.0 | No |
| | S (D1+D2)→T (D3) | 11.6 | 2.1 | 23.8 | 49.4 | 70.5 | 50.5 | | |
| 15 W | S (D1)→T (D2) | 8.2 | 1.3 | 18.5 | 50.5 | 55.1 | 51.9 | 5.0 | No |
| | S (D1+D2)→T (D3) | 6.7 | 1.1 | 18.3 | 51.2 | 56.0 | 51.7 | | |

To evaluate the adaptation cost in a real mobile environment, we conducted the WebQSP experiment described in Section 5.2 on the NVIDIA Jetson Orin Nano. As summarized in Table 17, adapting from D1 to D2 required only 14.2 Wh of energy—well within the capacity of current smartphones (e.g. the OnePlus 13 battery is rated at 22.92 Wh). Notably, this measurement was obtained on the large WebQSP D2 dataset containing 4,354,048 triples, so the energy demand for typical on-device adaptation is expected to be substantially lower. Extrapolating from these results, adapting 10,000 triples consumes roughly 0.05 Wh, which is less than 0.2% of a typical smartphone battery (based on the OnePlus 13). This extremely low adaptation cost indicates that MobileKGQA can be routinely retrained on-device as users interact with their devices, enabling the embedded LLM to incorporate new knowledge almost in real time.

## I.5 TRAINING COST FOR HASHING MODULE

Table 18: Changes in computation costs for training hashing module based on hashing dimensions.

| Dataset | Metric | dimension | | | |
|---------|--------|-----------|--------|--------|-------|
| | | 512bit | 256bit | 128bit | 64bit |
| WebQSP | Params(M) | 1.18 | 0.59 | 0.30 | 0.15 |
| | GFLOPs | 2.83 | 1.18 | 0.59 | 0.41 |
| | Time(s) | 16 | 15 | 15 | 14 |
| | VRAM(GB) | 2.60 | 2.38 | 2.35 | 2.33 |

Table 18 summarizes the computational resources required for training the hashing module with a 2304-dimensional embedding model. The results show that the training process is highly efficient in terms of resource usage, with all configurations completed in under 20 seconds on an NVIDIA Tesla V100 GPU. The low parameter count and computational cost indicate that the module requires minimal memory and processing power. Even with the largest hash dimension (512-bit), training completes within 16 seconds, and VRAM usage remains below 2.6 GB. Given these characteristics, our hashing module is well-suited for deployment in resource-constrained environments, including mobile devices, and can be easily integrated into real-time applications without significant computational overhead.

## I.6 IMPACT OF GENERATED QUERIES ON PERFORMANCE AND COMPUTATIONAL COST

To demonstrate that MobileKGQA can adapt to distribution shifts using only the computational resources available on mobile devices, we analyzed how performance and computational requirements change with the number of generated annotations. Table 19 presents the performance improvements observed when adapting MobileKGQA from D1 to D2 in WebQSP, based on the number of annotations generated by Gemma2 (2B) with a token generation speed of 40 tokens per second on mobile devices (OctoML, 2023). The results show that MobileKGQA requires just 35 minutes to generate 800 annotations on a mobile device, consuming less than 0.5 MB of storage. This demonstrates that

Table 19: Computational cost and performance after adaptation based on generated annotations.

| Dataset | Metric | Number of generated annotations | | | | |
|---------|--------|-----|------|------|------|------|
| | | **0** | **200** | **400** | **600** | **800** |
| WebQSP (D1→D2) | Token(K) | 0 | 14.0 | 22.9 | 38.3 | 50.9 |
| | F1† | 49.3 | 50.1 | 50.8 | 50.6 | 51.7 |
| | Time(min)‡ | 0 | 10.1 | 16.3 | 27.2 | 35.3 |
| | storage(MB) | 0 | 0.12 | 0.24 | 0.36 | 0.48 |

MobileKGQA is already feasible for mobile deployment. The F1 score was computed on the combined D1 + D2 dataset, and the inference time was measured on a mobile device at a processing rate of 40 tokens per second.

## I.7 QUALITY ANALYSIS ON GENERATED ANNOTATIONS

Table 20: Quality analysis on the generated annotations using various LLMs and WebQSP dataset. (red: best, blue: second-best)

| LLM | Generation Method | Metrics | | | |
|-----|-------------------|---------------|----------------|--------|------------|
| | | **ROUGE-L (%)** | **BERTScore (%)** | **Tokens** | **Time (min)** |
| Qwen 2 (0.5B, 4bit) | ReasoningLM's | 9.7 | 10.7 | 85220 | 13.7 |
| | Chain of Thought | 18.5 | 18.7 | **73847** | **12.2** |
| | MobileKGQA's | **30.2** | **33.8** | 116842 | 21.3 |
| Gemma 2 (2B, 4bit) | ReasoningLM's | 17.0 | 36.4 | **25401** | **6.8** |
| | Chain of Thought | 32.3 | 37.6 | 328687 | 53.2 |
| | MobileKGQA's | **42.8** | **48.7** | 97855 | 25.5 |
| Llama 3.1 (8B, 4bit) | ReasoningLM's | 12.9 | 26.3 | **29095** | **7.5** |
| | Chain of Thought | 31.2 | 37.7 | 540194 | 74.2 |
| | MobileKGQA's | **42.4** | **48.9** | 83742 | 24.2 |
| Phi 4 (14B, 4bit) | ReasoningLM's | 12.1 | 18.8 | **54045** | **16.9** |
| | Chain of Thought | 26.0 | 32.5 | 424289 | 118.2 |
| | MobileKGQA's | **41.9** | **48.4** | 89471 | 30.2 |

To evaluate the quality and efficiency of annotations generated by our method, we conducted an analysis based on four key metrics: ROUGE-L, BERTScore, number of generated tokens, and generation time. ROUGE-L and BERTScore assess how closely the generated questions resemble human-annotated questions, while the number of tokens and time reflect the generation efficiency. For this analysis, we utilized human-annotated questions from the WebQSP dataset. To generate annotations, relevant triples were first identified through SPARQL queries corresponding to each human-annotated question, and one triple combination was then randomly sampled for generation. Instances were excluded if appropriate triples could not be retrieved due to issues such as manual SPARQL queries and incorrect relation information. After this filtering process, a total of 2,101 annotations were evaluated. We compared our method against two baseline approaches—(1) ReasoningLM (RLM) (Jiang et al., 2023b), which employs few-shot prompting, and (2) Chain-of-Thought (CoT) (Wei et al., 2022), which enables step-by-step reasoning—across four different LLM settings, as reported in Table 20. For RLM, we used the prompt exactly as presented in the original paper, while for CoT, we utilized the prompt from Appendix L. All experiments were conducted on a NVIDIA Tesla V100 GPU.

Across all model configurations—from lightweight models like Qwen2 (0.5B, 4-bit) to larger ones such as Phi-4 (14B, 4-bit), which are still difficult to deploy on mobile devices—MobileKGQA consistently achieved the highest ROUGE-L and BERTScore. It also demonstrates that it requires only 28.4% of the tokens compared to CoT, which shows the second-best performance. When applying CoT to Qwen2-0.5B, we observed that the model often failed to follow the CoT prompt and instead produced short, dummy text. As a result, it generated fewer tokens and required less time than other on-device LLMs. These results demonstrate that our proposed method is both more effective and efficient than existing approaches. Furthermore, its robust performance across varying LLMs high-

lights its applicability not only to devices with more limited capabilities than today's smartphones, but also to future high-performance on-device LLMs.

## I.8 COMPARISON WITH OTHER HASHING METHODS

To evaluate the effectiveness of the proposed hashing module, we compared MobileKGQA against several alternative hashing method, including K-means–based quantization, PCA-based hashing, and locality-sensitive hashing (LSH). For a fair evaluation, we replaced only the hashing component while keeping all other parts of the MobileKGQA framework unchanged. Experiments were conducted on WebQSP and CWQ using 256-bit hash codes, and all reported scores are the mean ± standard deviation over three runs. As summarized in Table 21, MobileKGQA consistently outperforms all alternative hashing techniques.

Table 21: Performance Comparison on WebQSP and CWQ

| Method | WebQSP | CWQ |
|---|---|---|
| K-means Quantization | $67.9 \pm 0.05$ | $30.7 \pm 0.17$ |
| PCA | $65.5 \pm 0.16$ | $29.2 \pm 0.21$ |
| LSH | $68.2 \pm 0.08$ | $29.3 \pm 0.19$ |
| MobileKGQA | $\mathbf{69.4 \pm 0.08}$ | $\mathbf{33.3 \pm 0.18}$ |

To better understand these findings, we analyzed why the baseline hashing methods perform suboptimally in our setting:

**K-means Quantization**    Product quantization (PQ) is a widely used K-means–based method that compresses embeddings by partitioning a vector into multiple subvectors, each mapped to its nearest centroid. Because a global 256-bit codebook would require an exponentially large number of centroids, subvector partitioning is unavoidable for PQ. However, this partitioning fails to retain cross-subvector correlations—information that encodes important semantic dependencies learned by the reasoning module. As a result, retrieval performance degrades when PQ is used for hashing in KGQA.

**PCA-based Hashing**    PCA identifies directions of high variance but does not consider the binarization step that follows. When PCA outputs are thresholded to generate binary codes, substantial quantization errors occur, discarding semantically informative dimensions. This mismatch between projection and quantization leads to reduced retrieval quality in KGQA tasks, where semantically precise embeddings are crucial.

**Locality-Sensitive Hashing (LSH)**    LSH hashes embeddings using randomly sampled hyperplanes and preserves angular similarity in expectation. However, transformer embeddings encode semantic information along specific "privileged basis" directions (Team, 2023), and these data-dependent structures are further shaped by user-specific knowledge bases. Because LSH relies on distribution-agnostic random projections, it fails to capture such informative directions, resulting in suboptimal retrieval quality in our setting.

Overall, these findings demonstrate the advantage of MobileKGQA's learned hashing module. By training data-aware, task-specific transformations, it preserves semantic structure more effectively and reduces the quantization error inherent in traditional hashing approaches.

## I.9 ANALYSIS OF HASH COLLISIONS AND HASHING FAILURE CASES

We evaluate the robustness of the hashing module in MobileKGQA by analyzing both (i) the likelihood of hash collisions during retrieval and (ii) potential failure cases arising from insufficient hash capacity or distribution shifts.

**Collision Analysis.**    To assess whether hashing introduces ambiguity during retrieval, we measure collision rates across datasets with different graph densities and hop lengths. Table 22 reports the

collision rate together with mAP. For each query category, the collision rate and mAP were calculated as the probability that duplicated 256-bit hash codes appeared among the codes generated for retrieval. Across all settings, the collision rate remained **zero**, even in dense graphs such as MetaQA 3-hop and multi-hop reasoning tasks. This indicates that hash collisions are highly unlikely in the practical KGQA environments targeted by MobileKGQA.

This observation reflects a fundamental difference between KGQA hashing and conventional applications. While typical hashing (e.g., indexing, integrity checking) does not require reconstructing semantic information, KGQA specifically relies on hash representations to preserve semantic content for reasoning. This requirement forces KGQA hashing to retain richer information, making collisions significantly less likely in practice. For this reason, we adopt **mAP** as the primary metric, as it directly measures semantic preservation.

Table 22: Hash Collision Rates Across Graph Types, Densities, and Hop Lengths

| Graph Type | Density | # Hop | mAP | Collision Rate |
|---|---|---|---|---|
| WebQSP | 11.7 | 1 | 92.9 | 0 |
| | | 2 | 88.3 | 0 |
| CWQ | 53.7 | 1 | 92.5 | 0 |
| | | 2 | 87.7 | 0 |
| | | $\geqslant 3$ | 86.2 | 0 |
| MetaQA-1hop | 2548.8 | 1 | 89.9 | 0 |
| MetaQA-2hop | 3108.7 | 2 | 89.7 | 0 |
| MetaQA-3hop | 3141.7 | 3 | 89.7 | 0 |

**Failure Case Analysis.** In MobileKGQA, hashing failures may stem from two sources—insufficient hash capacity or distribution shift—but neither poses a meaningful challenge in practice. Regarding the first case, although retrieval performance could degrade if the data complexity exceeds the expressiveness of the hash space, our experiments on large and structurally rich benchmarks (WebQSP, CWQ) show that a compact 256-bit code can reliably handle millions of entities without any loss in performance. Furthermore, even when we intentionally reduced the hashing dimension to an extreme level (down to 16 bits) to induce hashing failures, the performance dropped by only 9.5%. These results suggest that capacity-related failures are unlikely to pose a practical concern in real-world settings.

Table 23: Performance Changes According to Hashing Dimension on WebQSP

| Metric | 256-bit | 128-bit | 64-bit | 32-bit | 16-bit |
|---|---|---|---|---|---|
| mAP | 0.907 | 0.877 | 0.827 | 0.690 | 0.567 |
| Hit (RM) | 77.5 | 76.6 | 75.5 | 72.3 | 70.1 (9.5% ↓) |

For the second failure mode, although distribution shifts may affect accuracy, our hashing module is extremely lightweight and can be retrained on-device with minimal overhead (Appendix I.5, Table 17), enabling rapid and efficient adaptation whenever the user's data distribution changes. As shown in the table below for WebQSP, the hashing module maintains stable performance throughout the entire adaptation process, demonstrating that retraining reliably preserves hashing quality under distribution shifts.

Table 24: Domain-wise Performance of the Hashing Module during Adaptation on WebQSP

| Metric | D1 | D1 + D2 | D1 + D2 + D3 |
|---|---|---|---|
| mAP | 0.913 | 0.881 | 0.893 |

## J  DETAILS ABOUT DOMAIN SPLIT

This section describes the details of how each domain is defined proposed to model distribution shifts. The questions in the dataset were embedded using an embedding model based on the Llama 3.1 8B model provided by Ollama, and the K-nearest neighbor algorithm was applied to cluster them into three distinct domains. Each domain was then randomly split into training (50%), validation (10%), and test (40%) sets. Since each dataset consists of queries and their associated graphs, splitting based solely on queries still ensures that the corresponding graphs are appropriately distributed across different domains. Table 25 presents the statistics of queries included in each domain, while Figures 3a and 3b illustrate the distribution of domain-specific queries in WebQSP, CWQ.

Table 25: Statistics of each domain split

| Dataset | Attribute | Domain1 | Domain2 | Domain3 |
|---|---|---|---|---|
| WebQSP | # question | 1596 | 1024 | 2080 |
|  | # train | 798 | 512 | 1040 |
|  | # validation | 159 | 102 | 208 |
|  | # test | 639 | 410 | 832 |
| CWQ | # question | 9601 | 13068 | 12020 |
|  | # train | 4800 | 6534 | 6010 |
|  | # validation | 960 | 1306 | 1202 |
|  | # test | 3841 | 5228 | 4808 |

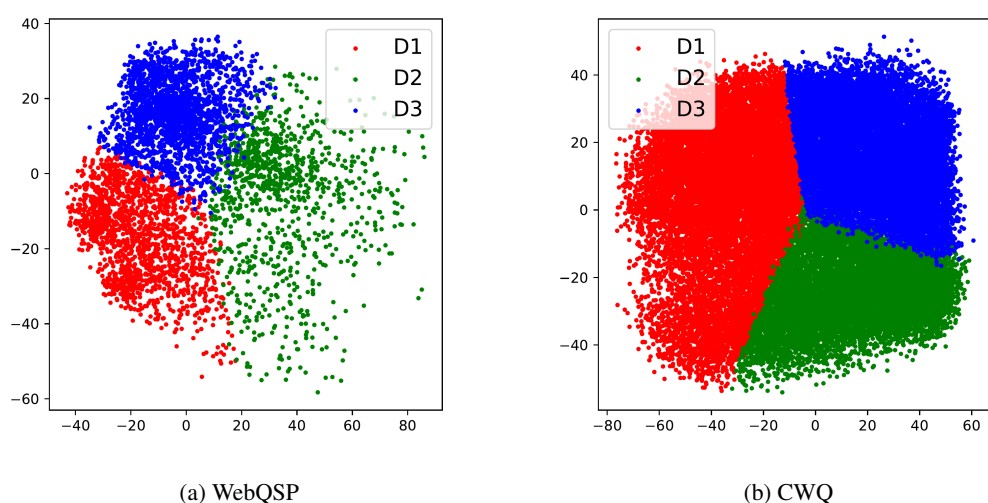

(a) WebQSP                    (b) CWQ

Figure 3: PCA visualization of question embeddings across different domains

## K    PROMPTS FOR MOBILEKGQA

---

**Answer Generation Prompt**

---

Reasoning Paths: {paths}
Please answer the question below by referencing the reasoning paths above. Construct your answer using the exact wording from the reasoning paths whenever possible. Provide all possible answers as a comma-separated list, outputting only the list.
Question: {question}

---

---

**Query Generation Prompt (Verbalization)**

---

Your task is to convert a single knowledge triple into a simple and grammatically correct English sentence.

Important instructions:
- Do NOT add any external knowledge or interpretation.
- Use the original words from the triple.
- Generate exactly one sentence.
- Do not include any additional explanation.

Example:
Triple: (Marfan syndrome, medicine.risk_factor.diseases, Myxomatous degeneration)
Sentence: Marfan syndrome is a risk factor for Myxomatous degeneration.
Triple: (Monotheism, religion.religion.beliefs, Sikhism)
Sentence: Monotheism is a belief of Sikhism.

Now perform the task with the following input:
Triple: {triples}

---

---

**Query Generation Prompt (Merge)**

---

You are a language model that integrates multiple factual sentences into a single English sentence that describes the answer, using only the provided information.

Instructions:
- Given input sentences, write one fluent and grammatically correct English sentence that describes the answer.
- Make sure all facts from the input sentences are included in the output sentence to describe the answer.
- Output only the final merged sentence. Do not provide any explanation or reasoning.
- If input sentences include encoded identifiers (e.g., m.0j2sdtx), replace it with an appropriate type-indicating phrase such as the person, the location, or the film, depending on the context.

### Example 1
Input Sentences:
- Lampedusa Airport is located in Italy.
- The capital of Italy is Rome.
- Italy was established on 2 June 1946.

Answer: Rome

Output: Rome is the capital of Italy, a country that was established on 2 June 1946 and where Lampedusa Airport is located.

### Example 2
Input Sentences:
- Gerald Baron Cohen is an accountant.
- Erran Baron Cohen is the parent of Gerald Baron Cohen.
- Erran Baron Cohen worked on the music for Grimsby.

Answer: Gerald Baron Cohen

Output: Gerald Baron Cohen is an accountant and the son of Erran Baron Cohen, who worked on the music for Grimsby.

### Example 3
Input Sentences:
- m.046csbd is the discoverer of the Euler–Lagrange equation.
- Mikhail Golovin was a student of m.046csbd.

Answer: Mikhail Golovin

Output: Mikhail Golovin was a student of the person who discovered the Euler–Lagrange equation.

Now perform the task with the following input:

Input Sentences:
{sentences}
Answer: {answer}

Output:

---

---

**Query Generation Prompt (Placeholder Generation)**

Your task is to classify a target phrase by replacing it with a type-indicating placeholder (e.g., "the person", "the record label", "the ideology") that best reflects its semantic category.

Instructions:
- You are given a target phrase and a sentence where it appears, which may provide helpful context.
- Your job is to determine the semantic type of the target phrase and replace it with an appropriate placeholder.
- Output only the type-indicating placeholder. Do not include any explanation.

### Example 1
Sentence: Kurt Gödel developed Gödel's completeness theorem.
Target phrase: Gödel's completeness theorem
placeholder: the theorem

### Example 2
Sentence: A&M Octone Records signed Dropping Daylight as one of their artists.
Target phrase: A&M Octone Records
placeholder: the record label

### Example 3
Sentence: David Wylie advocated for the issue of nonviolence.
Target phrase: nonviolence
placeholder: the ideology

### Example 4
Sentence: Gerald Baron Cohen is an accountant and the son of Erran Baron Cohen, who worked on the music for Grimsby.
Target phrase: Gerald Baron Cohen
placeholder: the person

Now perform the task with the following input:

Sentence: {merged_sentence}
Target phrase: {answer}
placeholder:

---

---

**Query Generation Prompt (Masking)**

---

Your task is to replace a specific phrase in a sentence with a type-indicating placeholder (e.g., "the person", "the record label", "the ideology") that semantically describes the original phrase.

Instructions:
- You are given a sentence and a target phrase that appears in the sentence.
- Replace the target phrase with a semantic placeholder phrase that accurately reflects its type (e.g., "the person" for a person's name, "the country" for a location, "the concept" for a theory).
- Output only type-indicating placeholder. Do not include any explanation or commentary.

### Example 1
Sentence: Kurt Gödel developed Gödel's completeness theorem.
Target phrase: Gödel's completeness theorem
Place holder: *the theorem*
Output: Kurt Gödel developed *the theorem*.

### Example 2
Sentence: A&M Octone Records signed Dropping Daylight as one of their artists.
Target phrase: A&M Octone Records
Place holder: *The record label*
Output: *The record label* signed Dropping Daylight as one of their artists.

### Example 3
Sentence: David Wylie advocated for the issue of nonviolence.
Target phrase: nonviolence
Place holder: *the ideology*
Output: David Wylie advocated for the issue of *the ideology*.

### Example 4
Sentence: Gerald Baron Cohen is an accountant and the son of Erran Baron Cohen, who worked on the music for Grimsby.
Target phrase: Gerald Baron Cohen
Place holder: *the person*
Output: *the person* is an accountant and the son of Erran Baron Cohen, who worked on the music for Grimsby.

Now perform the task with the following input:

Sentence: {merged}
Target phrase: {a_entity}
Place holder: {place_holder}
Output:

---

---

**Query Generation Prompt (Question Generation)**

---

Your task is to generate a natural-sounding question based on a sentence in which the answer has already been masked with a semantic placeholder (e.g., the person, the process, the river). You are also provided with one or more question entities, which must be included in the generated question.

Instructions:
- Convert the given sentence into a question where the part marked with asterisks is the answer.
- You must include question entities in the question.
- Output only the question without additional explanation.

### Example 1
Sentence: *the person*, an accountant, is the son of Erran Baron Cohen, who worked on the music for Grimsby.
Question entity: accountant, Grimsby
Question: Who is the accountant whose parent worked on the music for Grimsby?

### Example 2
Sentence: *the process* is the process through which plants convert sunlight into energy.
Question entity: sunlight, plants
Question: What is the process through which plants convert sunlight into energy?

### Example 3
Sentence: *the river* which flows through South America, is one of the longest rivers in the world.
Question entity: South America
Question: What is the longest river in the world that flows through South America?

Now perform the task with the following input:

Sentence: {masked}
Question entity: {question_entity}
Question:

---

---

**Query Generation Prompt (Refinement)**

---

Your task is to revise the given question to ensure that all specified question entities are explicitly included in it.
The original answer of the question, represented by the placeholder (e.g., the person, the process), must not change in the revised question.

Instructions:
• You are given a sentence containing a placeholder, a list of question entities, and a question.
• If any of the question entities are missing from the question, revise it to include them naturally and meaningfully.
• You must preserve the original answer implied by the placeholder. Do not change the type or identity of what the question is asking for.
• Your output should be the revised question only without explanation.

Example 1
Sentence: *the person*, who is an accountant, is the son of Erran Baron Cohen, who worked on the music for Grimsby.
Question entity: accountant, Grimsby
Question: Who is the person whose parent worked on the music?
Revised question: Who is the accountant whose parent worked on the music for Grimsby?

Example 2
Sentence: *the process* is the process through which plants convert sunlight into energy.
Question entity: sunlight, plants
Question: What is the process through which energy is produced?
Revised question: What is the process through which plants convert sunlight into energy?

Example 3
Sentence: *the river* which flows through South America, is one of the longest rivers in the world.
Question entity: South America
Question: What is one of the longest rivers in the world?
Revised question: What is the longest river in the world that flows through South America?

Example 4
Sentence: *the event* marked the beginning of the French Revolution.
Question entity: French Revolution
Question: What event marked the beginning of the revolution?
Revised question: What event marked the beginning of the French Revolution?

Now perform the task with the following input:

Sentence: {masked}
Question entity: {question_entity}
Question: {question}
Revised question:

---

## L    PROMPTS FOR ANNOTATION GENERATION USING CHAIN OF THOUGHT

---

**Query Generation Prompt (Chain of Thought)**

---

Given triples, create a question that illustrates relationships between multiple entities, ensuring that the question entity is explicitly mentioned in the question and that the answer entity is the correct answer to the generated question.

Follow these steps:

1. Find a triple containing the Answer Entity. Based on this triple, form an initial question asking about the Answer Entity.
2. Check if there are other triples involving the Answer Entity. If found, incorporate relevant information into the question to make it more specific.
3. If there are no more triples involving the Answer Entity, select a triple that contains any entity already mentioned in the current question. Add its information to further enrich the question.
4. Continue enriching the question following step 3 until no relevant triples remain.

Example:

Given Triples:
- Accountant - people.person.profession - Gerald Baron Cohen
- Gerald Baron Cohen - people.person.parents - Erran Baron Cohen
- Erran Baron Cohen - film.film.music - Grimsby
Question Entity:
- Accountant, Grimsby
Answer Entity:
- Gerald Baron Cohen
Generation Steps:
1. Identify a triple containing the answer entity:
- Triple: Accountant - people.person.profession - Gerald Baron Cohen
- Question: "Who is the accountant?"
2. Check for additional triples involving answer entity:
- Triple: Gerald Baron Cohen - people.person.parents - Erran Baron Cohen
- Question: "Who is the accountant whose parent is Erran Baron Cohen?"
3. Check other triples already mentioned in the current question:
- Triple: Erran Baron Cohen - film.film.music - Grimsby
- Question: "Who is the accountant whose parent composed the music for the film Grimsby?"
4. No more triples to process. Finalize the generation process

Now perform the task with the following input:

Given Triples: {triples}
Question Entity: {question_entity}
Answer Entity: {answer_entity}
Output format:
Generation Steps:
(Step-by-step analysis showing which triples are used and how the question is enriched)
Question: (generated question)

---

## M  ANNOTATION GENERATION EXAMPLES

| Generation Steps | Example |
|---|---|
| Sampled Information | **triples:**
1. KITT - film.film_character.portrayed_in_films - Knight Rider 2000
2. Knight Rider 2000 - film.performance.actor - William Daniels
3. Knight Rider 2000 - film.performance.special_performance_type - Voice
**question entity:** KITT
**answer entity:** William Daniels |
| Verbalization | **sentences:**
1. KITT was portrayed in films as a character from Knight Rider 2000.
2. William Daniels performed in Knight Rider 2000.
3. Knight Rider 2000 features a voice performance. |
| Merge | **merged sentence about the answer:**
William Daniels, who played the character KITT in Knight Rider 2000, featured a voice performance in the film. |
| Placeholder Generation | **Placeholder for the answer:** the person |
| Masking | **masked sentence:**
*the person*, who played the character KITT in Knight Rider 2000, featured a voice performance in the film. |
| Question Generation | **generated question:**
Who featured a voice performance in the film as KITT in Knight Rider 2000? |
| Refinement | Since the question entity KITT is already included in the question, the refinement step is skipped. |

| Generation Steps | Example |
|---|---|
| Sampled Information | **triples:**
1. General Motors - automotive.company.make_s - Vauxhall
2. Vauxhall - common.topic.notable_types - Automobile Make
**question entity:** General Motors
**answer entity:** Vauxhall |
| Verbalization | **sentences:**
1. General Motors makes Vauxhall vehicles.
2. Vauxhall is a notable type of automobile make. |
| Merge | **merged sentence about the answer:**
Vauxhall is a notable type of automobile make made by General Motors. |
| Placeholder Generation | **Placeholder for the answer:** the automobile make |
| Masking | **masked sentence:**
*the automobile make* is a notable type of automobile make made by general motors. |
| Question Generation | **generated question:**
what is the notable type of automobile make made by general motors? |
| Refinement | Since the question entity general motors is already included in the question, the refinement step is skipped. |

