# OpenReview forum: "MobileKGQA: On-Device KGQA System on Dynamic Mobile Environments"
_ICLR.cc/2026/Conference — ICLR 2026 Poster_

### Official Review · Reviewer_RLPG · 2025-10-28

**Soundness:** 3
**Presentation:** 3
**Contribution:** 3
**Rating:** 6
**Confidence:** 4

**Summary:**

This paper addresses the critical challenges of deploying Knowledge Graph Question Answering (KGQA) systems on mobile devices, specifically the significant obstacles posed by resource constraints and continuous data accumulation. The authors propose MobileKGQA, the first on-device KGQA system specifically designed to adapt to evolving knowledge graphs with minimal resource requirements. The system employs embedding hashing to compress high-dimensional embeddings into compact binary codes, significantly reducing computational and storage overhead. It also introduces a novel annotation generation method based on sequential reasoning to handle distribution shifts caused by newly accumulated data. Extensive experiments on the NVIDIA Jetson Orin Nano platform demonstrate that MobileKGQA achieves 20.3% higher performance while consuming only 30.4% of the energy of the state-of-the-art (SOTA) model. On standard KGQA benchmarks, it uses only 7.2% of the computation and 9% of the parameters of the SOTA model while maintaining comparable performance.

**Strengths:**

This paper demonstrates significant strengths across several key dimensions, presenting a compelling case for its contribution to the field of on-device AI and knowledge graph reasoning.
- This work exhibits high originality by tackling the relatively unexplored problem of on-device training and adaptation for KGQA systems, not just on-device inference. It creatively combines ideas from different domains: Embedding Hashing for KGQA, Stepwise Annotation Generation for Adaptation
- The study is executed with high quality and methodological rigor. The experimental evaluation is comprehensive and convincing.
- The paper is generally well-structured and clearly written. The logical presentation of the three-phase pipeline (Hashing, Retrieval, Adaptation) is clear. Figures 1 and 2 effectively illustrate the core workflows. The use of tables to summarize comparative results and ablation studies is clear and facilitates comparison.
- This work is highly important for both academic research and practical applications. It directly addresses the growing need for privacy-preserving, efficient, and adaptive AI on personal devices. By enabling on-device training, it avoids data privacy concerns associated with cloud-based model retraining. The proposed techniques pave the way for a new generation of mobile systems that can continuously learn from user data without compromising performance or battery life. The provided prompts and methodology serve as a valuable resource for subsequent research in the community.

**Weaknesses:**

- Insufficient Analysis of Hashing Module Limitations:

The hashing module is central to MobileKGQA's efficiency claims, but the paper lacks a deeper analysis of its inherent limitations – particularly regarding hash collisions. Compressing high-dimensional embeddings into low-dimensional binary codes is inherently a lossy process. The paper does not discuss the possibility of different entities or relations being mapped to the same or very similar hash codes (i.e., hash collisions) and their potential impact. This weakens the claim of the system's robustness. If the collision rate increases as the knowledge graph expands, it could contaminate the answer candidate set, leading to degraded retrieval accuracy and final QA performance.

- Limited Diversity in Experimental Datasets Affects Generality Claims:

The paper's experiments are validated only on two datasets (WebQSP and CWQ) derived from Freebase. To claim broad applicability, testing on knowledge graphs with different structural characteristics is necessary. For example, graphs like Wikidata are denser with more complex relations, while some domain-specific graphs might be sparser. The performance of the hashing module might vary with graph properties. In sparse graphs, with less semantic information, the discriminative power of hash codes might be insufficient; in dense graphs, with high semantic overlap, the risk of hash collisions increases. The current experimental setup cannot demonstrate the system's effectiveness across these diverse scenarios.

- Lack of Comparison with Simpler Hashing or Dimensionality Reduction Baselines:

The paper fails to adequately compare its proposed mutual information maximization-based hashing method against simpler, more established baseline techniques. There is no comparison against classical dimensionality reduction/hashing techniques like Locality-Sensitive Hashing (LSH) or Principal Component Analysis (PCA). It is therefore difficult to ascertain to what extent the performance gains stem from the paper's sophisticated hashing objective function versus merely the act of "using hashing" itself. This makes it challenging to accurately assess the innovative contribution of this module.

- Imprecisions in Result Presentation:

In Table 2a, GNN-RAG's disk usage is reported as 0.54GB, while MobileKGQA's is 0.6GB. However, the narrative in the main text might lead readers to perceive MobileKGQA as optimal across all dimensions.

**Questions:**

- Regarding Hash Collisions: Did you evaluate the hash collision rate? If collisions exist, how does the system mitigate their potential negative impact on the retrieval of multi-hop reasoning paths?
- Regarding Generalization Capability: Do you have plans, or could you provide results, extending your evaluation to knowledge graphs beyond Freebase (e.g., Wikidata)? For dense graphs where semantic relations highly overlap, does your method require specific adjustments or optimizations?
- Regarding Method Comparison: Could you provide comparative results against simpler hashing baselines (e.g., LSH)? This would help the community better understand the added value brought by the "maximizing mutual information" design choice.

---

> ### Author Response · Authors · 2025-11-19
> **Official Comment by Authors (1/3)**
>
> **[Q1&W1]  Regarding Hash Collisions: Did you evaluate the hash collision rate? If collisions exist, how does the system mitigate their potential negative impact on the retrieval of multi-hop reasoning paths?**
>
>
> Following the suggestions from reviewers i4Rc and RLPG, we first conducted **experiments on hash collisions** and then analyzed **potential hashing failure cases.**
>
> (1) Hash Collision Experiments
>
> **Collision rate across datasets with varying graph densities and hop lengths**
>
> | Dataset       | Density  | # Hop | mAP   | Collision Rate |
> |---------------|----------|-------|-------|----------------|
> | WebQSP        | 11.7     | 1     | 92.9  | 0              |
> |               |          | 2     | 88.3  | 0              |
> | CWQ           | 53.7     | 1     | 92.5  | 0              |
> |               |          | 2     | 87.7  | 0              |
> |               |          | >3    | 86.2  | 0              |
> | MetaQA-1hop   | 2548.8   | 1     | 89.9  | 0              |
> | MetaQA-2hop   | 3108.7   | 2     | 89.7  | 0              |
> | MetaQA-3hop   | 3141.7   | 3     | 89.7  | 0              |
>
> To analyze the hash collision rate of the proposed hashing method, we evaluated its behavior across different graph densities and query hop counts. For each query category, the collision rate was calculated as the probability that duplicated 256-bit hash codes appeared among the codes generated for retrieval. **Across a wide range of graph densities and query hop counts, the collision rate remained zero,** indicating that hash collisions are highly unlikely to occur in the practical graph database environments targeted by MobileKGQA, **even in multi-hop reasoning cases.** This outcome stems from a fundamental difference between KGQA and conventional uses of hash codes. In typical applications such as indexing or integrity checking, hash codes are not required to support information reconstruction. In contrast, KGQA must recover and directly utilize the semantic information encoded in the hash, necessitating the preservation of much more of the original signal. As a result, KGQA hash codes inherently retain far richer information than conventional hash codes, making collisions extremely rare in practice. Accordingly, we adopt mAP as the primary evaluation metric, as it more directly reflects semantic preservation.
>
> (2) Hashing Failure Case
> In MobileKGQA, hashing failures may stem from two sources—**insufficient hash capacity** or **distribution shift**—but neither poses a meaningful challenge in practice.
> **First,** although retrieval performance could degrade if the data complexity exceeds the expressiveness of the hash space, our experiments on large and structurally rich benchmarks (WebQSP, CWQ) show that **a compact 256-bit code can reliably handle millions of entities** without any loss in performance. Furthermore, even when we intentionally reduced the hashing dimension to an extreme level (down to 16 bits) to induce hashing failures, the performance dropped by only 9.5%. These results suggest that capacity-related failures are unlikely to pose a practical concern in real-world settings.
>
>
> **Changes in Performance according to the hashing dimension in WebQSP (RM: Reasoning Module)**
> | Metric | 256-bit | 128-bit | 64-bit | 32-bit | 16-bit |
> |--------|---------|---------|--------|--------|--------|
> | mAP |  0.907 | 0.877 | 0.827 |  0.690 | 0.567 |
> | Hit(RM)    |  77.5 | 76.6 | 75.5 | 72.3 | 70.1 (9.5% drop)|
>
>
> **Second,** although distribution shifts may affect accuracy, our hashing module is extremely lightweight and can be retrained on-device with minimal overhead (Appendix I.5, Table 17), enabling rapid and efficient adaptation whenever the user’s data distribution changes. As shown in the table below for WebQSP, **the hashing module maintains stable performance throughout the entire adaptation process,** demonstrating that retraining reliably preserves hashing quality under distribution shifts.
>
> **Domain-wise performance of the hashing module on WebQSP**
> | Metric | D1     | D1 + D2 | D1+ D2 + D3 |
> |--------|---------|---------|--------|
> | mAP |  0.913 | 0.881 | 0.893 |

---

> ### Author Response · Authors · 2025-11-19
> **Official Comment by Authors (2/3)**
>
> **[Q2&W2] Regarding Generalization Capability: Do you have plans, or could you provide results, extending your evaluation to knowledge graphs beyond Freebase (e.g., Wikidata)? For dense graphs where semantic relations highly overlap, does your method require specific adjustments or optimizations?**
>
>
> We thank the reviewer for raising this valuable point regarding the generalizability of our approach to knowledge graphs beyond Freebase. While extending the evaluation to Wikidata is a natural direction, it is not appropriate for our study because Wikidata contains a large proportion of encrypted entities (machine identifiers) without human-readable names, making it not well-suited for LLMs to effectively reason over them. To address the reviewer’s concern in a meaningful way, we instead conducted additional experiments on the MetaQA datasets, which are not Freebase-based and are built upon the MovieQA [1] knowledge base. As shown in the table below, MetaQA datasets exhibit much denser graph structures than WebQSP, CWQ, and even Wikidata [2], providing a more rigorous environment to assess our model’s performance under high-density semantic relations.
>
> **Triples per Entity Across Datasets**
> | Dataset        | # Triples per Entity |
> |----------------|----------------------|
> | WebQSP         | 11.7                 |
> | CWQ            | 53.7                 |
> | Wikidata       | 187.4                |
> | MetaQA-1hop    | 2548.8               |
> | MetaQA-2hop    | 3108.7               |
> | MetaQA-3hop    | 3141.7               |
>
> The table below summarizes the performance of our model and the GNN-RAG baseline on these MetaQA datasets. **Across all datasets, mobileKGQA and GNN-RAG achieve comparable performance, demonstrating that our hashing module remains robust even on graphs that are approximately 60–300× denser than WebQSP and CWQ.** This indicates strong generalization to highly dense semantic structures. Moreover, combined with the experimental results reported in Q1 that hash collisions remain extremely rare across a wide range of graph densities, these results provide consistent evidence of mobileKGQA’s robustness and generalization ability.
>
> **F1 Score on MetaQA Datasets**
> | Model       | MetaQA-1hop      | MetaQA-2hop      | MetaQA-3hop      |
> |-------------|------------------|------------------|------------------|
> | GNN-RAG     | 95.9 ± 0.53      | 96.7 ± 0.78      | 88.7 ± 0.86       |
> | mobileKGQA  | 95.8 ± 0.68      | 96.7 ± 0.63      | 88.6 ± 0.92       |
>
> [1] Tapaswi, M., Zhu, Y., Stiefelhagen, R., Torralba, A., Urtasun, R., & Fidler, S. (2016). MovieQA: Understanding Stories in Movies through Question-Answering. arXiv [Cs.CV]. Retrieved from http://arxiv.org/abs/1512.02902
>
> [2] Haller, A., Polleres, A., Dobriy, D., Ferranti, N., & Rodríguez Méndez, S. J. (2022). An Analysis of Links in Wikidata. The Semantic Web: 19th International Conference, ESWC 2022, Hersonissos, Crete, Greece, May 29 – June 2, 2022, Proceedings, 21–38. Presented at the Hersonissos, Greece. doi:10.1007/978-3-031-06981-9_2

---

> ### Author Response · Authors · 2025-11-19
> **Official Comment by Authors (3/3)**
>
> **[Q3&W3] Regarding Method Comparison: Could you provide comparative results against simpler hashing baselines (e.g., LSH)? This would help the community better understand the added value brought by the "maximizing mutual information" design choice.**
>
> We thank reviewers DB7Z and RLPG for highlighting the need to compare our hashing module with simpler alternatives. Following the suggestion, we replaced our hashing module with these methods on WebQSP and CWQ, keeping all other components of MobileKGQA unchanged. As shown in the Table below, MobileKGQA consistently outperforms all alternatives. Results are reported as mean ± standard deviation over three runs.
>
> F1-score of the reasoning module under different hashing modules (all evaluated with 256 bit hash codes)
> | Hashing Method        | WebQSP             | CWQ               |
> |-----------------------|--------------------|-------------------|
> | K-means Quantization  | 67.9 ± 0.05        | 30.7 ± 0.17       |
> | PCA-based Hashing     | 65.5 ± 0.16        | 29.2 ± 0.21       |
> | Locality-Sensitive Hashing (LSH) | 68.2 ± 0.08 | 29.3 ± 0.19       |
> | **MobileKGQA** | **69.4 ± 0.08**    | **33.3 ± 0.18**   |
>
> To explain these results, we outline the reasons why the baseline methods underperform in our setting below.
>
> (1) K-means Quantization.
> Product quantization splits embeddings into subvectors and quantizes each independently, since using a global (ex. 256 bit) codebook is computationally infeasible. However, this partitioning destroys cross-subvector correlations that encode important semantic structure, leading to degraded retrieval accuracy.
>
> (2) PCA-based Hashing.
> PCA maximizes variance but is not designed for subsequent binarization. The binarization step introduces large quantization errors, discarding essential semantic information needed for retrieval.
>
> (3) Locality-Sensitive Hashing (LSH).
> LSH uses random hyperplanes that are not learned. However, transformer embeddings encode semantics along meaningful “privileged” directions [1], and user-specific data distributions might exist in certain axes. Therefore, LSH fails to reflect critical semantic variations, resulting in suboptimal performance.
>
> [1] Anthropic Transformer Circuits Thread (https://transformer-circuits.pub/2023/privileged-basis/index.html)
>
> **[W4]  In Table 2a, GNN-RAG's disk usage is reported as 0.54GB, while MobileKGQA's is 0.6GB. However, the narrative in the main text might lead readers to perceive MobileKGQA as optimal across all dimensions.**
>
> We appreciate the reviewer for pointing out this subtle inconsistency in our description, which we had overlooked. Following the reviewer’s suggestion, we corrected the statement accordingly. Specifically, on Line 316, we revised the sentence to:
>
> “Furthermore, it (MobileKGQA) is the second most optimal model in terms of memory and storage efficiency.”

---

> ### Author Response · Authors · 2025-11-26
> **Request for Feedback on Revisions**
>
> Dear Reviewer RLPG,
>
> Thank you very much for your detailed and constructive review. We have now addressed your concerns by (1) providing deeper empirical analysis of hashing limitations and collision risks, (2) evaluating our method on denser non-Freebase datasets (MetaQA), (3) adding comparisons against simpler hashing baselines (e.g., LSH, PCA, K-means), and (4) correcting the description regarding storage usage.
>
> We hope these revisions sufficiently resolve the issues you raised and help clarify the contribution of our work. If they do, we would greatly appreciate your consideration in updating the review. Please let us know if there are any remaining questions or additional clarifications we can provide.
>
> Thank you again for your thoughtful feedback and valuable time.

---

### Official Review · Reviewer_i4Rc · 2025-11-01

**Soundness:** 3
**Presentation:** 3
**Contribution:** 3
**Rating:** 6
**Confidence:** 3

**Summary:**

The paper proposes MobileKGQA, the first KGQA system that supports both inference and on-device training under strict resource constraints. To reduce memory and computation, the authors introduce a hashing module that compresses high-dimensional embeddings into compact binary codes while preserving semantic similarity. To adapt to continuously evolving knowledge graphs without cloud retraining, the paper further introduces a stepwise annotation generation method that constructs question–answer pairs locally. Experiments on WebQSP and CWQ show that MobileKGQA achieves performance comparable to state-of-the-art lightweight KGQA systems while using only 7.2% of computation and 9% of parameters.

**Strengths:**

S1: The paper presents the first KGQA system that supports on-device training and adaptation to evolving knowledge graphs, demonstrating clear practical value for privacy-sensitive mobile environments.

S2: The proposed hashing module achieves large reductions in computation and storage while preserving semantic similarity, leading to near-SOTA performance with only a small fraction of the parameters and energy consumption. Real-device experiments further strengthen the credibility of the claims.

**Weaknesses:**

W1: The theoretical justification of the hashing module relies on simplifying assumptions (e.g., independence between embedding norm and orientation), and the paper lacks deeper empirical analysis of potential failure cases, such as hash collisions or cold-start relations with limited semantic context.

W2: Although the system claims full on-device adaptation, it relies on pre-trained embedding models to encode newly added triples. The paper does not analyze whether these embedding models can run efficiently on mobile hardware, making it unclear if the entire pipeline is truly on-device in practical scenarios.

**Questions:**

Q1: Can the proposed system also adapt when triples are removed from the KG, or is the current adaptation mechanism designed only for newly added knowledge?

Q2: Is there any scenario where hashing leads to ambiguous collisions that harm reasoning? Are there failure cases or examples?

---

> ### Author Response · Authors · 2025-11-19
> **Official Comment by Authors (1/3)**
>
> **[Q1]  Can the proposed system also adapt when triples are removed from the KG, or is the current adaptation mechanism designed only for newly added knowledge?**
>
> We appreciate the reviewer’s insightful question. **As discussed in Appendix I.2 (Table 15), we also conducted experiments simulating data deletion scenarios, where certain triples were removed from the knowledge graph.** The results consistently demonstrate that our proposed system outperforms GNN-RAG under such conditions as well.
>
> Table 15. Hit scores of GNN-RAG and MobileKGQA on the WebQSP dataset under varying data deletion ratios. (MobileKGQA used 256-bit hash codes. *: p-value <0.01, **: p-value <0.001.)
>
> | Deletion Ratio | GNN-RAG (F1)      | MobileKGQA (F1)      |
> |----------------|-------------------|-----------------------|
> | 20%            | 84.2 ± 0.2        | 87.2 ± 0.4*           |
> | 40%            | 77.2 ± 0.6        | 84.9 ± 0.8**          |
> | 60%            | 70.0 ± 0.2        | 83.3 ± 0.6**          |
> | 80%            | 56.7 ± 0.3        | 71.4 ± 0.3**          |
>
> Compared with data accumulation, data deletion is more tractable because the validity of annotations can be easily checked through simple rule-based queries (e.g., SPARQL). In addition, our annotation generation process provides a reasoning path, question, and answer for each annotation, which makes consistency verification straightforward. Based on your suggestion, we have updated the manuscript to explicitly state that our framework supports both data accumulation and data deletion.

---

> ### Author Response · Authors · 2025-11-19
> **Official Comment by Authors (2/3)**
>
> **[Q2&W1]  Is there any scenario where hashing leads to ambiguous collisions that harm reasoning? Are there failure cases or examples?**
>
>
> Following the suggestions from reviewers i4Rc and RLPG, we first conducted **experiments on hash collisions** and then analyzed **potential hashing failure cases.**
>
> (1) Hash Collision Experiments
>
> **Collision rate across datasets with varying graph densities and hop lengths**
>
> | Dataset       | Density  | # Hop | mAP   | Collision Rate |
> |---------------|----------|-------|-------|----------------|
> | WebQSP        | 11.7     | 1     | 92.9  | 0              |
> |               |          | 2     | 88.3  | 0              |
> | CWQ           | 53.7     | 1     | 92.5  | 0              |
> |               |          | 2     | 87.7  | 0              |
> |               |          | >3    | 86.2  | 0              |
> | MetaQA-1hop   | 2548.8   | 1     | 89.9  | 0              |
> | MetaQA-2hop   | 3108.7   | 2     | 89.7  | 0              |
> | MetaQA-3hop   | 3141.7   | 3     | 89.7  | 0              |
>
>
> To analyze the hash collision rate of the proposed hashing method, we evaluated its behavior across different graph densities and query hop counts. For each query category, the collision rate was calculated as the probability that duplicated 256-bit hash codes appeared among the codes generated for retrieval. **Across a wide range of graph densities and query hop counts, the collision rate remained zero,** indicating that hash collisions are highly unlikely to occur in the practical graph database environments targeted by MobileKGQA. This outcome stems from a fundamental difference between KGQA and conventional uses of hash codes. In typical applications such as indexing or integrity checking, hash codes are not required to support information reconstruction. In contrast, KGQA must recover and directly utilize the semantic information encoded in the hash, necessitating the preservation of much more of the original signal. As a result, KGQA hash codes inherently retain far richer information than conventional hash codes, making collisions extremely rare in practice. Accordingly, we adopt mAP as the primary evaluation metric, as it more directly reflects semantic preservation.
>
> (2) Hashing Failure Case
> In MobileKGQA, hashing failures may stem from two sources—**insufficient hash capacity** or **distribution shift**—but neither poses a meaningful challenge in practice.
> **First,** although retrieval performance could degrade if the data complexity exceeds the expressiveness of the hash space, our experiments on large and structurally rich benchmarks (WebQSP, CWQ) show that **a compact 256-bit code can reliably handle millions of entities** without any loss in performance. Furthermore, even when we intentionally reduced the hashing dimension to an extreme level (down to 16 bits) to induce hashing failures, the performance dropped by only 9.5%. These results suggest that capacity-related failures are unlikely to pose a practical concern in real-world settings.
>
>
> **Changes in Performance according to the hashing dimension in WebQSP (RM: Reasoning Module)**
> | Metric | 256-bit | 128-bit | 64-bit | 32-bit | 16-bit |
> |--------|---------|---------|--------|--------|--------|
> | mAP |  0.907 | 0.877 | 0.827 |  0.690 | 0.567 |
> | Hit(RM)    |  77.5 | 76.6 | 75.5 | 72.3 | 70.1 (9.5% drop)|
>
>
> **Second,** although distribution shifts may affect accuracy, our hashing module is extremely lightweight and can be retrained on-device with minimal overhead (Appendix I.5, Table 17), enabling rapid and efficient adaptation whenever the user’s data distribution changes. As shown in the table below for WebQSP, **the hashing module maintains stable performance throughout the entire adaptation process,** demonstrating that retraining reliably preserves hashing quality under distribution shifts.
>
> **Domain-wise performance of the hashing module on WebQSP**
> | Metric | D1     | D1 + D2 | D1+ D2 + D3 |
> |--------|---------|---------|--------|
> | mAP |  0.913 | 0.881 | 0.893 |

---

> ### Author Response · Authors · 2025-11-19
> **Official Comment by Authors (3/3)**
>
> **[W2]   Although the system claims full on-device adaptation, it relies on pre-trained embedding models to encode newly added triples. The paper does not analyze whether these embedding models can run efficiently on mobile hardware, making it unclear if the entire pipeline is truly on-device in practical scenarios.**
>
> As the reviewer correctly points out, the quality of the pretrained embedding model across diverse datasets is indeed crucial to the performance of our proposed method. **However,  there are two complementary research directions that support full on-device operation of embedding models:**
>
> **(1) Compact embedding models with strong generalization**
>
> A growing body of work—including GTE [1], E5 [2], and the BGE family [3]—demonstrates strong performance on MTEB benchmarks while offering lightweight variants suitable for edge deployment. Many of these models range from tens to a few hundred million parameters, a scale that can already be executed efficiently on mobile hardware. These findings indicate that high-quality embeddings can be generated directly on-device.
>
> **(2) Decoder-only LLMs used as embedding models.**
> Recent studies further show that decoder-only LLMs can serve as effective embedding generators while retaining their generative capabilities (e.g., LLM2Vec [4], GEM [5]). Such approaches allow a single LLM to support both response generation and embedding, implying that, in the near future, dedicated embedding models may no longer be required at all.
>
> In summary, **(1) efficient embedding models already exist that can run on mobile devices today,** and **(2) near-term trends indicate that on-device LLMs will soon be capable of generating embeddings themselves,** enabling the entire pipeline to operate effectively on mobile hardware.
>
>
> [1] Li, Z., Zhang, X., Zhang, Y., Long, D., Xie, P., & Zhang, M. (2023). Towards General Text Embeddings with Multi-stage Contrastive Learning. arXiv [Cs.CL]. Retrieved from http://arxiv.org/abs/2308.03281
>
> [2] Wang, L., Yang, N., Huang, X., Yang, L., Majumder, R., & Wei, F. (2024). Multilingual E5 Text Embeddings: A Technical Report. arXiv Preprint arXiv:2402. 05672.
>
> [3] Chen, J., Xiao, S., Zhang, P., Luo, K., Lian, D., & Liu, Z. (2024). BGE M3-Embedding: Multi-Lingual, Multi-Functionality, Multi-Granularity Text Embeddings Through Self-Knowledge Distillation. arXiv [Cs.CL]. Retrieved from http://arxiv.org/abs/2402.03216
>
> [4] BehnamGhader, P., Adlakha, V., Mosbach, M., Bahdanau, D., Chapados, N., & Reddy, S. (2024). LLM2Vec: Large Language Models Are Secretly Powerful Text Encoders. First Conference on Language Modeling. Retrieved from https://openreview.net/forum?id=IW1PR7vEBf
>
> [5] Zhang, C., Zhang, Q., Li, K., Nuthalapati, S. V., Zhang, B., Liu, J., … Fan, X. (2025). GEM: Empowering LLM for both Embedding Generation and Language Understanding. arXiv [Cs.CL]. Retrieved from http://arxiv.org/abs/2506.04344

---

> ### Author Response · Authors · 2025-11-26
> **Request for Feedback on Revisions**
>
> Dear Reviewer i4Rc,
>
> Thank you once again for your valuable and constructive comments. In response, we have (1) expanded our analysis of potential hashing failures and collision risks, (2) clarified that our adaptation mechanism supports both data addition and deletion, and (3) elaborated on the practicality of running embedding models fully on-device.
>
> We hope these updates adequately address your concerns and help strengthen the contribution of our work. If they do, we would greatly appreciate your consideration in revising the review. Please let us know if there are any remaining questions or additional clarifications we can provide.
>
> Thank you very much for your time and insightful feedback.

---

### Official Review · Reviewer_DB7Z · 2025-11-06

**Soundness:** 3
**Presentation:** 3
**Contribution:** 3
**Rating:** 6
**Confidence:** 4

**Summary:**

The paper introduces MobileKGQA, an on-device KGQA system that trains and runs fully on mobile/edge hardware. It combines a hashing-based retriever (compressing LLM embeddings to binary codes) with a lightweight reasoning module, plus a stepwise on-device annotation generation to handle evolving KGs under tight compute/privacy constraints. On WebQSP/CWQ and a Jetson Orin Nano, it reports comparable SOTA accuracy while using much less params, and on-device tests beats baselines.

**Strengths:**

1. This paper is studying a practical and important problem, i.e., to make the on device KGQA.
2. Although the reviewer does not think the proposed the algorithm is very novel, these algorithms are sound and supported by theoretical verification and solves the problem nicely.
3. This paper has thorough and solid empirical evidence to support the effectiveness of the proposed method. Especially, this paper really runs the algorithm on a Jetson, which is more practical.

**Weaknesses:**

1. For the hashing algorithm, this seems to work similar as quantization to binary with cosine constraints, so the reviewer is wondering if a simple kmeans quantization can work better?
2. The privacy preserving is an important aspect the paper is asserting, is there any privacy leakage when the LLM are generating the final answer?

**Questions:**

See weaknesses.

---

> ### Author Response · Authors · 2025-11-19
> **Official Comment by Authors**
>
> **[W1]  For the hashing algorithm, this seems to work similar as quantization to binary with cosine constraints, so the reviewer is wondering if a simple kmeans quantization can work better?**
>
> We thank reviewers DB7Z and RLPG for highlighting the need to compare our hashing module with simpler alternatives. Following the suggestion, we replaced our hashing module with other hashing methods, keeping all other components of MobileKGQA unchanged. As shown in the Table below, MobileKGQA consistently outperforms all alternatives. Results are reported as mean ± standard deviation over three runs.
>
> F1-score of the reasoning module under different hashing modules (all evaluated with 256 bit hash codes)
> | Hashing Method        | WebQSP             | CWQ               |
> |-----------------------|--------------------|-------------------|
> | K-means Quantization  | 67.9 ± 0.05        | 30.7 ± 0.17       |
> | PCA-based Hashing     | 65.5 ± 0.16        | 29.2 ± 0.21       |
> | Locality-Sensitive Hashing (LSH) | 68.2 ± 0.08 | 29.3 ± 0.19       |
> | **MobileKGQA** | **69.4 ± 0.08**    | **33.3 ± 0.18**   |
>
> To explain these results, we outline the reasons why the baseline methods underperform in our setting below.
>
> (1) K-means Quantization.
> Product quantization splits embeddings into subvectors and quantizes each independently, since using a global (ex. 256 bit) codebook is computationally infeasible. However, this partitioning destroys cross-subvector correlations that encode important semantic structure, leading to degraded retrieval accuracy.
>
> (2) PCA-based Hashing.
> PCA maximizes variance but is not designed for subsequent binarization. The binarization step introduces large quantization errors, discarding essential semantic information needed for retrieval.
>
> (3) Locality-Sensitive Hashing (LSH).
> LSH uses random hyperplanes that are not learned. However, transformer embeddings encode semantics along meaningful “privileged” directions [1], and user-specific data distributions might exist in certain axes. Therefore, LSH fails to reflect critical semantic variations, resulting in suboptimal performance.
>
> [1] Anthropic Transformer Circuits Thread (https://transformer-circuits.pub/2023/privileged-basis/index.html)
>
>
>
> **[W2]  The privacy preserving is an important aspect the paper is asserting, is there any privacy leakage when the LLM are generating the final answer?**
>
> In our current design, all core processes — hashing, reasoning, answer generation, and adaptation — are executed entirely on the device. The on-device LLM only accesses locally stored, pre-processed data, and no intermediate outputs or queries leave the device. Consequently, assuming proper device-level security, the risk of data leakage beyond the device boundary is significantly minimized.

---

> ### Author Response · Authors · 2025-11-26
> **Request for Feedback on Revisions**
>
> Dear Reviewer DB7Z,
>
> Thank you again for your constructive feedback. We have carefully addressed your concerns by (1) adding comparisons with K-means, PCA-based hashing, and LSH, and (2) providing a clearer description of how our method ensures privacy.
>
> We hope these revisions adequately address the points you raised and further strengthen our contribution. If they do, we would greatly appreciate your consideration in updating the review. Please let us know if there are any remaining questions or additional clarifications we can provide.
>
> Thank you very much for your time and valuable insights.

---

### Author Response · Authors · 2025-11-21
**Response to All Reviewers**

We sincerely appreciate all of the reviewers’ insightful comments and helpful suggestions. We have incorporated the requested analyses and experimental results into the revised manuscript, and the updated content is marked in blue.

---

> ### Author Response · Authors · 2025-12-01
> **minor revision of the manuscript**
>
> We have uploaded a minor revised version with a clearer description for Lines 234–236.

---

### Author Response · Authors · 2025-12-03
**Author message for AC summarizing the rebuttal**

Dear AC,

We are grateful for your time and for guiding the review process under such exceptional conditions. We appreciate the opportunity to provide this final message to support your evaluation.

---

**Reviewer-Identified Strengths**
- First KGQA system to support on-device training and adaptation, offering high practical value for privacy-preserving, efficient mobile AI.
- Sound algorithms creatively combine Embedding Hashing and Stepwise Annotation Generation.
- Thorough empirical evidence supports effectiveness, including execution on a real device.
- The hashing module achieves large reductions in storage and computation, delivering near-SOTA performance with minimal resources.
- The paper is well-structured and clearly written, logically presenting the three-phase pipeline with effective visual aids.

---

**Reviewer-Identified Concerns and Rebuttals**

**1. Hash Collision & Failure Case Analysis**: Reviewer i4Rc (Q2&W1) / Reviewer RLPG (Q1&W1)
Collision rates were zero across all graph densities and hop lengths, showing that 256-bit hashes are highly robust. Even extreme compression caused only mild degradation, and the lightweight hashing module can be retrained on-device to handle distribution shifts.

**2. Comparison with Other Hashing Methods**: Reviewer DB7Z (W1) / Reviewer RLPG (Q3&W3)
Experiments with K-means-based method, PCA hashing, and LSH show that MobileKGQA consistently outperforms all simpler baselines, confirming the value of the proposed hashing approach.

**3. Applicability to Data Deletion Scenarios**: Reviewer i4Rc (Q1)
Under simulated triple deletion (20–80%), MobileKGQA consistently outperformed GNN-RAG. The proposed method supports both data accumulation and data deletion.

**4. Generalization Beyond Freebase**: Reviewer RLPG (Q2&W2)
Evaluation on the much-denser MetaQA datasets (non-Freebase) shows performance on par with GNN-RAG, demonstrating strong generalization to structurally different and denser KGs.

**5. Privacy Leakage in On-Device Answer Generation**: Reviewer DB7Z (W2)
All components operate entirely on-device, and the LLM only accesses locally processed data. No information leaves the device, minimizing privacy risks.

**6. Practicality of Running Embedding Models On-Device**: Reviewer i4Rc (W2)
Efficient embedding models already exist that can run on mobile devices today, and near-term trends indicate that on-device LLMs will soon be capable of generating embeddings themselves, enabling the entire pipeline to operate effectively on mobile hardware.

---

In summary, we believe our rebuttal and new analyses resolve all reviewer concerns and strengthen both the technical validity and practical value of MobileKGQA. We appreciate your time and hope this overview supports your decision.

Sincerely,
The Authors

---

### Meta-Review · Area_Chair_kcc3 · 2025-12-27

**Summary:**

The paper introduces MobileKGQA, an on-device KGQA system that trains and runs fully on mobile/edge hardware. It combines a hashing-based retriever with a lightweight reasoning module, plus a stepwise on-device annotation generation to handle evolving KGs under tight compute/privacy constraints. It received scores of 666. Reviewers agree that the proposed method is sound, and the empirical experiments are thorough, including execution on a real device. The raised concerns are also well addressed by the authors. Therefore, the AC would like to recommend acceptance.

**Reviewer Concerns:**

Concerns adequately addressed:

1. Hash collision & failure case analysis by Reviewer i4Rc and RLPG.

2. Comparison with other hashing methods by Reviewer DB7Z and RLPG: the authors added results on K-means-based method, PCA hashing, and LSH.

3. Applicability to data deletion scenarios by Reviewer i4Rc;

4. Generalization beyond freebase by Reviewer RLPG: the authors added results on MetaQA.


Concerns insufficiently addressed:

1. The experimental scope could be strengthened by including results on larger-scale benchmarks to better demonstrate the generalization capability of the method.

**Reviewer Scores:**

The reviewers are positive about the paper; however, the AC is not sure whether the reviewers are open to further increase the scores.

---

### Decision · Program_Chairs · 2026-01-26

Accept (Poster)